# The Response of Tropical Cyclone Intensity to Changes in Environmental Temperature

James M. Done[1,2,*] Gary M. Lackmann[3,*] , Andreas F. Prein[1]

[1]National Center for Atmospheric Research, 3090 Center Green Drive, Boulder, Colorado 80301, USA
[2]Willis Research Network, 51 Lime St, London, EC3M 7DQ, UK
[3]Department of Marine, Earth and Atmospheric Sciences, North Carolina State University, Raleigh, North Carolina 27607, USA
* These authors contributed equally to this work.

*Correspondence to*: James M. Done (done@ucar.edu)

**Abstract.** Theory indicates that tropical cyclone intensity should respond to environmental temperature changes near the surface and in the tropical cyclone outflow layer. While the sensitivity of tropical cyclone intensity to sea surface temperature is well understood, less is known about the role of upper-level stratification. In this paper, we combine historical data analysis and idealised modelling to explore the extent to which historical low-level warming and upper-level stratification can explain observed trends in the tropical cyclone intensity distribution. Observations and modelling agree that historical global environmental temperature changes coincide with higher lifetime maximum intensities. Observations suggest the response depends on the tropical cyclone intensity itself. Hurricane-strength storms have intensified at twice the rate of weaker storms per unit surface and upper tropospheric warming, and we find faster warming of low-level temperatures in hurricane environments than the tropical mean. Idealized simulations respond in the expected sense to various imposed changes in the near-surface temperature and upper-level stratification representing present-day and end-of-century thermal profiles and agree with tropical cyclones operating as heat engines. Removing upper tropospheric warming or stratospheric cooling from end-of-century experiments results in much smaller changes in potential intensity or realized intensity than between present-day and end-of-century. A larger proportional change in thermodynamic disequilibrium compared to thermodynamic efficiency in our simulations suggests that disequilibrium, not efficiency, is responsible for much of the intensity increase from present-day to end-of-century. The limited change in efficiency is attributable to nearly constant outflow temperature in the simulated TCs among the experiments. Observed sensitivities are generally larger than modelled sensitivities, suggesting that observed tropical cyclone intensity change responds to a combination of the temperature change and other environmental factors.

**Non-Technical Summary.** We know that warm oceans generally favour TC activity. Less is known about the role of air temperature above the oceans and extending into the lower stratosphere. Our analysis of historical records and computer simulations suggests that TCs strengthen in response to historical temperature change while also being influenced by other environmental factors. Ocean warming drives much of the strengthening, with changes in the efficiency of TC heat transfer contributing very little.

## 1 Introduction

Understanding how tropical cyclones (TCs) and their impacts respond to climate change is of critical scientific and societal importance (e.g., Knutson et al., 2020). However, TC response to environmental change is complex and multi-faceted. Here, we use observations and idealized models to examine the TC intensity response to changes in the environmental near-surface and upper-level temperatures.

Historical global surface temperature trend analyses show significant warming since the mid-1970s, attributed to anthropogenic forcing (Meehl et al., 2004; 2012). Yet trends in the vertical thermal structure and their attribution are less well understood (O'Gorman and Singh, 2013; Prein et al., 2017). Since the mid-1970s most datasets show that the troposphere has warmed while the lower stratosphere has cooled (e.g., Thompson et al., 2012; Philipona et al., 2018). However, analysing these trends is particularly challenging in the global tropics because of sparse long-term historical upper-air records and the potential for artificial trends driven by observing system changes (e.g., Thorne et al., 2011). Indeed, Vecchi et al. (2013) showed marked differences in the magnitude of the thermal changes among a collection of observational and reanalysis datasets.

Uncertainty in temperature trends also arises from the complexity of the driving mechanisms and their representation in reanalyses (Emanuel et al., 2013; Vecchi et al., 2013) and general circulation models (GCMs). A historical warming maximum in the upper troposphere can be explained through moist adiabatic ascent above warming oceans and has been attributed to increasing greenhouse gas forcing (Santer et al., 2005; 2008). A shift in the moist adiabat corresponds to larger warming aloft than at the surface. For the lower stratosphere, a strengthened Brewer-Dobson circulation has been proposed as a mechanism contributing to the cooling (Butchart, 2014). Here, cooling occurs through enhanced adiabatic cooling and reduced ozone concentration due the to upwelling of ozone-poor tropospheric air. At the same time, observed step changes in cooling have been attributed to the volcanic eruptions of El Chichón in 1982 and Mt. Pinatubo in 1991 (Fujiwara et al., 2015). Ramaswamy et al. (2006) isolated the role of changes in ozone, carbon dioxide, aerosols, and solar radiation in observed lower stratospheric cooling, concluding that anthropogenic factors were the driver of overall cooling between the late 1970s and the early 2000s.

The representation of these complex mechanisms differs among GCMs and may contribute to the wide range in the magnitude of GCM-simulated profile changes (Cordero and Forster, 2006; Santer et al., 2008; Gettelman et al., 2010; Hill and Lackmann, 2011; Hardiman et al., 2014). GCMs are generally unable to reproduce observed profile change at the uppermost tropospheric levels (Po-Chedley and Fu, 2012; Mitchell et al., 2013), though whether this is due to model or observational error remains unclear. This large spread among models and disagreement with observations may limit our ability to project tropical cyclone (TC) intensity. Emanuel et al. (2013) conclude that tropopause layer cooling contributed to increased TC potential intensity in the North Atlantic basin and that improved process representation of profile changes in GCMs is critically needed to improve TC projections.


As the thermal profile has changed, so has the distribution of global TC intensity (e.g., Kossin et al., 2013; Sobel et al., 2016).
A recent analysis of a homogeneous historical TC intensity record from 1979 to 2017 revealed a statistically robust increase
in global lifetime maximum intensity (Kossin et al., 2020). The observed intensity distribution has not simply shifted to higher
intensities, but has become increasingly bimodal (Holland and Bruyère, 2014; Lee et al., 2016; Jewson and Lewis, 2020).

These changes in the TC intensity distribution may be attributable to a variety of environmental and internal processes,
including both natural and anthropogenic effects. Changes in vertical wind shear (Ting et al., 2019), humidity (Dai, 2006),
temperature (at the sea surface, near surface, and in the TC outflow layer), and the nature of incipient disturbances may all
contribute to TC intensity change. It is also understood that the observational datasets used in these analyses have limitations
(e.g., Landsea et al., 2006; Klotzbach and Landsea, 2015), although recent efforts have reduced these uncertainties (e.g.,
Knutson et al., 2019; Kossin et al., 2020; Emanuel, 2021). TC intensity sensitivity to the underlying sea surface temperature
(SST), or more accurately the thermal disequilibrium between the SST and the near-surface atmosphere, is relatively well
understood (Emanuel, 1987; Elsner et al., 2008; Strazzo et al., 2015; Gilford et al. 2017). Global average TC intensity scales
by 2.5% per degree Kelvin SST warming (Knutson et al., 2019). Yet the magnitude and mechanistic response of TC intensity
to changes in upper-level stratification and TC outflow layer temperatures are less well understood.

A Carnot heat engine has been used to link TC intensity with near-surface and TC outflow layer temperatures (Emanuel, 1986;
1991; 2006; Ramsay, 2013; Pauluis and Zhang, 2017). This maximum potential intensity (PI) theory suggests that TC intensity
changes in response to SSTs that drive atmosphere-ocean disequilibrium and to the engine's efficiency (the temperature
difference between the surface and the level of the TC outflow) (e.g., Emanuel 1988; Holland 1997). Specifically, the square
of PI is proportional to the product of the thermodynamic efficiency and the thermodynamic disequilibrium. Changes in
disequilibrium, rather than efficiency, have been shown to dominate PI variations for seasonal variations (Gilford et al., 2017)
and interannual to decadal variations (Rousseau-Rizzi and Emanuel, 2021). In idealised axisymmetric simulations under
radiative-convective equilibrium, PI increased by about 1 $ms^{-1}$ per degree of lower stratospheric cooling, and by about 1.5 to
2 $ms^{-1}$ per degree of surface warming (Ramsay, 2013). But the relative importance of disequilibrium and efficiency likely
varies by basin (Gilford et al. 2017). SST and outflow temperature are strongly linked when the outflow is confined to the
troposphere thereby limiting TC intensification associated with ocean warming (Shen et al., 2000; Hill and Lackmann, 2011;
Tuleya et al., 2016). However, there is greater potential for larger efficiency changes when the outflow extends above the
tropopause and occurs in the cooling lower stratosphere.

The realized response of the TCs themselves may be quite different from the response of PI (e.g., Vecchi et al., 2013). This
could be due to the different TC outflow layer temperatures in the PI algorithm versus the actual storm. But perhaps more
important are environmental factors such as wind shear and humidity acting in combination with internal processes such as
asymmetries in the distribution of moist entropy (Riemer et al. 2010; Alland et al. 2021a,b; Wadler et al. 2021) or in the
distribution of convection (Rogers et al. 2013; Zawislak et al. 2016; Alvey et al. 2020) that can limit the TC intensity response.
Furthermore, the realized response of TCs appears to depend on the TC intensity itself. Indeed, the highest sensitivity to surface
warming resides in the strongest storms (e.g., Elsner et al., 2008; Knutson et al., 2010).

We hypothesize that observed environmental temperature changes exert predictable influences on TC intensity. Furthermore,
we explore whether historic near-surface and upper-level temperature changes are sufficient to explain past trends in the TC
intensity distribution. Our approach blends historical data analysis with idealized numerical modelling. Observational analyses
bring together a global homogenized radiosonde temperature dataset with a homogeneous TC intensity record to minimize
contamination by artificial trends. Naturally, observed trends in TC intensity are not due to changes in temperature alone and
respond to changes in other environmental factors. Our goal is to isolate the influence of temperature change on TC intensity.
We focus on a global-scale analysis over a 37-year historical period - scales at which TC intensity should be more strongly
constrained by thermodynamic change than by other environmental or geographic factors (Deser et al., 2012). Idealized
numerical modelling further isolates and quantifies the TC intensity response to observed trends and future changes in
environmental temperatures.

The next section describes the observation datasets and analysis procedures, and the numerical model experiments. Results of
the observational analysis and idealized numerical model experiments are presented in Sect. 3. A synthesis and concluding
discussion is provided in Sect. 4.
**2 Methods**
**2.1 Historical temperature and tropical cyclone datasets**
We use multiple temperature and TC datasets to characterise historical trends and the relationships between TC intensity and
thermal structure. Temperature data are compared across radiosonde soundings and two reanalysis datasets and related to two
historical TC datasets.

Global radiosonde data are obtained from the Radiosonde Observation Correction Using Reanalyses (RAOBCORE) v1.5.1,
available on a 10° × 5° grid, 16 pressure levels, and twice daily (Haimberger, 2007; Haimberger et al., 2012). RAOBCORE
was developed to be suitable for climate applications and was created by applying a time-series homogenization to the
Integrated Global Radiosonde Archive (IGRA; Durre et al., 2006). This procedure uses temperature differences between
radiosonde observations and background forecasts from the European Centre for Medium-Range Weather Forecasts
(ECMWF) Re-Analysis (ERA-40, Uppala et al., 2005) to correct discontinuities tied to observing system changes and remove
persistent biases. These corrections are particularly important for lower stratospheric temperatures where measurements are
susceptible to radiation errors (Sherwood et al., 2005). Haimburger et al. (2008) showed that RAOBCORE compares
favourably with satellite-derived estimates of temperature trends in the upper troposphere and lower stratosphere consistent
with theoretical and model expectations. Sounding profiles are sufficiently numerous to characterise the thermal structure from
the 925-hPa level up to 50 hPa. While sounding locations in TC genesis regions are sparse, their spatial representativeness for
temperature scales with the large radius of deformation at low latitudes. In addition, we only use stations that have at least 70
% complete records over the period 1981 to 2017 and do not contain breakpoints. Breakpoints are detected following the
methods described in Prein and Heymsfield (2020). Briefly, four different breakpoint detection algorithms are applied and
time series for which more than two algorithms identified a breakpoint in the same year were excluded.
The two reanalysis datasets analysed here, both produced by the ECMWF, are the Interim reanalysis (ERA-I; Dee et al., 2011;
accessed from European Centre for Medium-Range Weather Forecasts, 2009) and the more recent ERA5 (Hersbach et al.,
2020; accessed from European Centre for Medium-Range Weather Forecasts, 2019). These reanalyses differ in important ways
that may affect trends in near-surface temperatures and upper-level stratification, including horizontal and vertical grid spacing,
model physics, data assimilation technique, and the data sources assimilated. The horizontal grid spacings are 79 km/TL255
(ERA-I) and 31 km/TL639 (ERA5), and the numbers of vertical levels and vertical extent are 60 levels up to 10 hPa for ERA-
I and 137 levels up to 1 hPa for ERA5.
ERA-I and ERA5 assimilate vast quantities of *in situ*, radiosonde, and remote sensing observations, and the observing systems
change over time. This can lead to discontinuities in the simulated time series (Dee et al., 2011; Simmons et al., 2014). ERA-
I assimilates the RAOBCORE data and ERA5 assimilates radiosonde data that have been homogenized using a newer
procedure that uses neighbouring stations rather than departure statistics alone. ERA5 contains a pronounced cold bias in the
lower stratosphere from 2000 to 2006 due to the use of inappropriate background error covariances (Hersbach et al., 2020;
Simmons et al., 2020). This bias has been corrected in ERA5.1 which is a rerun of ERA5 for the period 2000-2006 only
(Simmons et al., 2020; accessed from European Centre for Medium-Range Weather Forecasts, 2020). For our analysis we join
ERA5 and ERA5.1 by replacing ERA5 with ERA5.1 for the years 2000 to 2006 and continue to refer to this merged dataset
as ERA5.
Observations of historical TCs are taken from two sources: The International Best Track Archive for Climate Stewardship
version 4 (IBTrACS, Knapp et al., 2010, downloaded on June 14, 2021) and a reanalysed intensity record provided by Kossin
et al. (2020). The IBTrACS has formed the basis for many studies of TC variability and change. Here, we use USA agency
data, which are largely derived from the National Hurricane Center's HURricane DATa 2nd generation (HURDAT2) dataset
and reports from the Joint Typhoon Warning Center. However, spatial and temporal variations in the instrumental observing
system challenge the interpretation of TC variability and change, particularly in the early record (e.g., Landsea et al., 2006;
Klotzbach and Landsea, 2015). Indeed, substantial differences across the reporting agencies (Knapp and Kruk, 2010) can
contaminate global climatologies (Schreck et al., 2014). In response, Kossin et al. (2013) reanalysed the historical intensity
record by applying an intensity algorithm (the advanced Dvorak Technique, ADT) to a homogenized geostationary satellite
dataset (the Hurricane Satellite record, HURSAT). The resulting ADT-HURSAT dataset was recently extended to cover the
period 1979 to 2017 (Kossin et al., 2020). The key advantage of ADT-HURSAT compared to IBTrACS is its consistency in
time and space which makes it suitable for trend analysis, especially from 1981 onwards. Both TC datasets are included here
to demonstrate the sensitivity of TC intensity change to artifacts of the datasets, and to connect results back to prior work.

The 37-year observational analysis period of 1981 to 2017 is chosen as a balance between data availability and to roughly
coincide with the start of the recent warming trend (e.g., Rahmstorf et al., 2017, their Fig. 2) and its influence on global TC
behaviour (Holland and Bruyère, 2014).

## 2.2 Idealized model experiments

We hypothesize that observed environmental temperature changes exert predictable influences on trends in the intensification
rate and maximum intensity of TCs. As discussed above, previous studies have explored the sensitivity of TC intensity to both
the tropical upper-tropospheric warming maximum and lower stratospheric cooling. Changes in temperature stratification near
the tropopause may influence the sensitivity of TC outflow temperature for a given SST warming (and therefore also influence
the thermodynamic efficiency). We use ensembles of simulations from an axisymmetric model to test these predictions and
quantify the magnitude of these influences on TC intensity.

The axisymmetric TC capability of Cloud Model 1 (CM1, Bryan and Fritsch, 2002; Bryan and Rotunno, 2009a) is well suited
for our experiments. The limitations of axisymmetric simulations are outweighed by the reduced computational expense,
which allows us to run ensembles of simulations. Axisymmetric models have proven useful in the evaluation of TC maximum
intensity (e.g., Rotunno and Emanuel, 1987; Bryan and Rotunno, 2009a; Hakim, 2011; Rousseau-Rizzi and Emanuel, 2019).
We acknowledge that some three-dimensional effects, such as vortex Rossby waves, are known to be important to TC intensity
(e.g., Wang, 2002; Gentry and Lackmann, 2010; Persing et al., 2013). So too are asymmetric thermodynamic processes such
as downdrafts and radial ventilation that can occur as a response to TC-environment interactions. While axisymmetric models
miss the component of the TC response due to internal thermodynamic and kinematic asymmetries, they offer a controlled
experimental design to start to link theory and observations. Thus, the response of axisymmetric vortices to changes in the
thermodynamic profile is deemed sufficient to test our hypotheses, but fully 3-dimensional simulations are needed to
investigate this limitation. The axisymmetric domain in our simulations features a 4 km grid length, a model top of 25 km (59
vertical levels), and a radial domain length of 1500 km. At radial distances greater than 280 km the grid length stretches to the
larger grid spacing. Sensitivity tests to a doubling of the radial domain length and a simultaneous doubling of the radial distance
at which the grid length stretches showed the sensitivity is small compared to changes in physics options or responses to
temperature changes (not shown). The horizontal mixing length in this version of CM1 is a linear function of surface pressure,
varying from 100 m at 1015 hPa to 1000 m at 900 hPa (Bryan, 2012).

We initialize CM1 (version r19.10) with the Dunion (2011) "moist tropical" sounding, derived from western North Atlantic
rawinsonde data from 1995 to 2002 (Fig. 1a). The model is initialized with a weak vortex (~12 ms$^{-1}$ maximum azimuthal
velocity in gradient thermal wind balance) like that in the control simulation of Rotunno and Emanuel (1987). A potentially
important difference between our experimental design and that of Rotunno and Emanuel (1987) is that our initial conditions
are not in a state of radiative-convective equilibrium. This is to assess the influence of temperature profile differences more
directly during the TC intensification stage, although we acknowledge that the TC begins to modify the environment
immediately, and we have not eliminated these changes in our simulations. Our present-day simulations feature an SST of
28°C, close to the near-surface air temperature (following Bryan and Rotunno 2009b)

We ran the simulations for 8 days, which allowed the idealized TCs to intensify to a maximum and then equilibrate to a quasi-
steady-state intensity. We recognize that much longer integrations have been used in several equilibrium studies (e.g., Hakim,
2011; Ramsay, 2013), but TC modification of the environment in longer integrations would limit our ability to detect
environmental influences. Shorter simulations also limit the effect of excessive large-scale drying in the subsidence region
leading to storm weakening found in some longer CM1 simulations (Rousseau-Rizzi et al., 2021). Given our goal of examining
TC responses to changes in environmental temperatures, we focus on the core steady-state (CS) period where intensity varies
only slowly after the time of peak core strength (Rousseau-Rizzi et al., 2021), though we also present the peak core strength
given its approximate equivalence to LMI. Owing to the sensitivity of simulated TC intensity to various model
parameterization choices, we ran an ensemble of 21 simulations for each environmental profile, varying the turbulence,
radiation, sea surface, and microphysical parameterizations (Tables 1, and A1).


**Table 1: CM1 model physics ensemble namelist choices for the surface model (sfcmodel), ocean model (oceanmodel), surface**
**exchange coefficients (isftcflx), atmospheric radiation (radopt), relaxation term that mimics atmospheric radiation (rterm), and**
**explicit moisture scheme (ptype); see Table A1 for specific settings for each of the 21 ensemble members.**

| parameter | description |
| --- | --- |
| sfcmodel | CM1 (1), "WRF" (2), "revised WRF" (3), GFDL (4), MYNN (6) |
| oceanmodel | constant SST (1), ocean mixed layer model (2) |
| isftcflx | Donelan (1), or Donelan/Garratt for Cd and Ce (2) |

| radopt | simple (0, with rterm = 1), NASA (1), or RRTMG (2) |
| ptype | Morrison (5) or Thompson (3) |

To explore the sensitivity of simulated TC intensity to changes in the environmental thermodynamic profile, we ran five additional 21-member ensemble experiments (Table 2). These were primarily designed to explore TC intensity response to extrapolated observational trends based on RAOBCORE data discussed in Sect. 2.1 and presented in Sect. 3.1. The "mid-century" experiment corresponds to conditions approximately in the year 2050 if current trends are extrapolated, and the "end-of-century" experiment applies changes extrapolated over a century-long period (Fig. 1c). SSTs for the mid- and end-of-century experiments were chosen to be close to the near-surface air temperature. Two additional experiments allow us to isolate the sensitivity of TC intensity to specific changes observed in tropical temperature profiles. The "no upper warming maximum" ensemble is based on a temperature change profile that is nearly constant with height in the troposphere (Fig. 1d), and the "no stratospheric cooling" simulations explore the TC response to a temperature change profile that eliminates lower stratospheric cooling (Fig. 1e). Recognizing the limitations in the extrapolation of current observational trends, we ran an additional ensemble experiment based on a multi-model mean of IPCC AR5 GCM tropical change profiles, for end-of-century conditions under the RCP8.5 scenario (Fig. 1b, and see Jung and Lackmann, 2019, their Table 2). For all simulations involving temperature perturbations, relative humidity is held constant, resulting in increased water vapor content with warming. This assumption is supported by observations (e.g., Dai 2006; Willett et al. 2007) in addition to theoretical and modelling studies (e.g., Allen and Ingram 2002; Held and Soden 2006; Pall et al. 2007).

**Table 2: Ensemble experiments and maximum intensity (i.e., $P_{min}$); values are for time-filtered time series. For three right columns, numbers in parentheses represent standard deviation. A Butterworth low-pass time filter was applied to remove high-frequency fluctuations. Core steady-state (CS) $P_{min}$ is taken over simulation hours 150 to 193, while $P_{min}$ is peak intensity. "Complex" denotes the 13-member ensemble subset with complex radiation parameterization. Settings for the Emanuel potential intensity (E-PI) calculation, based on the pyPI software package (Gilford, 2021), include dissipative heating (Bister and Emanuel, 1998), an enthalpy-drag coefficient ratio of 0.9, and a wind reduction coefficient of 0.9.**

| Experiment | SST | E-PI | $P_{min}$ (full ensemble) | $P_{min}$ (complex) | CS $P_{min}$ (complex) |
|---|---|---|---|---|---|
| Present-day | 301.2 K (28.0 °C) | 923.4 hPa (74.7 ms$^{-1}$) | 917.8 hPa (10.8 hPa) | 913.3 hPa (8.7 hPa) | 920.5 hPa (10.9 hPa) |

| | | | | | |
|---|---|---|---|---|---|
| Mid-Century | 301.8 K (28.6 °C) | 920.1 hPa (75.7 ms$^{-1}$) | 913.7 hPa (12.0 hPa) | 912.1 hPa (9.8 hPa) | 917.2 hPa (13.7 hPa) |
| End of Century | 302.4 K (29.2 °C) | 917.1 hPa (76.4 ms$^{-1}$) | 907.0 hPa (10.3 hPa) | 906.0 hPa (8.5 hPa) | 913.3 hPa (10.5 hPa) |
| No upper warming max | 302.4 K (29.2 °C) | 916.4 hPa (76.4 ms$^{-1}$) | 909.0 hPa (11.6 hPa) | 906.8 hPa (10.5 hPa) | 911.0 hPa (13.7 hPa) |
| No stratos. cooling | 302.4 K (29.2 °C) | 917.1 hPa (76.4 ms$^{-1}$) | 909.5 hPa (12.0 hPa) | 906.5 hPa (8.8 hPa) | 916.2 hPa (13.3 hPa) |
| GCM RCP 8.5 | 304.5 K (31.3 °C) | 910.9 hPa (77.5 ms$^{-1}$) | 903.5 hPa (12.8 hPa) | 901.0 hPa (10.2 hPa) | 908.1 hPa (12.9 hPa) |



Despite temporal variability, the ensemble mean intensity appears close to the analytical value predicted by the Emanuel (1988)
maximum potential intensity (E-PI, Table 2); we recognize that considerable uncertainty also exists in the E-PI values owing
to various choices that go into that calculation. We also note that the E-PI algorithm used here is formulated using a Convective
Available Potential Energy (CAPE)-based definition of E-PI, which does not depend explicitly on efficiency and
disequilibrium. Rather, it is based on the equivalence between disequilibrium and the difference between environmental CAPE
and saturation CAPE. Rousseau-Rizzi et al. (2022) show that the two formulations are physically linked via parcels' surface
moist static energy, thus increasing confidence in our use of the CAPE-based formulation.

Based on the thermodynamic and Carnot efficiency considerations mentioned in Sect. 1 and the E-PI calculations shown in
Table 2, we predict *a priori* that the present-day simulation would produce the weakest ensemble-mean TC, followed in order
of increasing intensity by the mid-century and end-of-century simulations. We further expect that simulations omitting the
tropical upper warming maximum would be slightly stronger than the default end-of-century ensemble and that the ensemble
removing stratospheric cooling would be slightly weaker in intensity relative to the default end-of-century run. We expect the
GCM-based ensemble to yield the strongest storm, given significantly greater warming. Of course, the numerical simulations
are not constrained to agree with these theoretically motivated predictions.

To further test our hypotheses relating changes in TC intensity to environmental temperature changes, we computed
thermodynamic efficiency and thermodynamic disequilibrium following Emanuel (1987; 1988) and Gilford (2021). Given the
availability of high-resolution numerical simulations, we also computed the simulated TC outflow temperature directly,
defined as the temperature of air with outward radial flow exceeding $1.0\ \mathrm{ms^{-1}}$ and cloud ice mixing ratio exceeding $10^{-5}\ \mathrm{kg\ kg^{-1}}$
. Experimentation with these threshold values demonstrates that this setting works well to represent the temperature of the
cirrostratus outflow layer, though the ensemble average values obtained were not highly sensitive to changes in the radial
velocity or cloud ice mixing ratio thresholds (not shown). In our analysis of derived outflow temperatures, we noted substantial
differences between simulations conducted with "complex" versus "simple" representations of radiation and have stratified
the results accordingly.

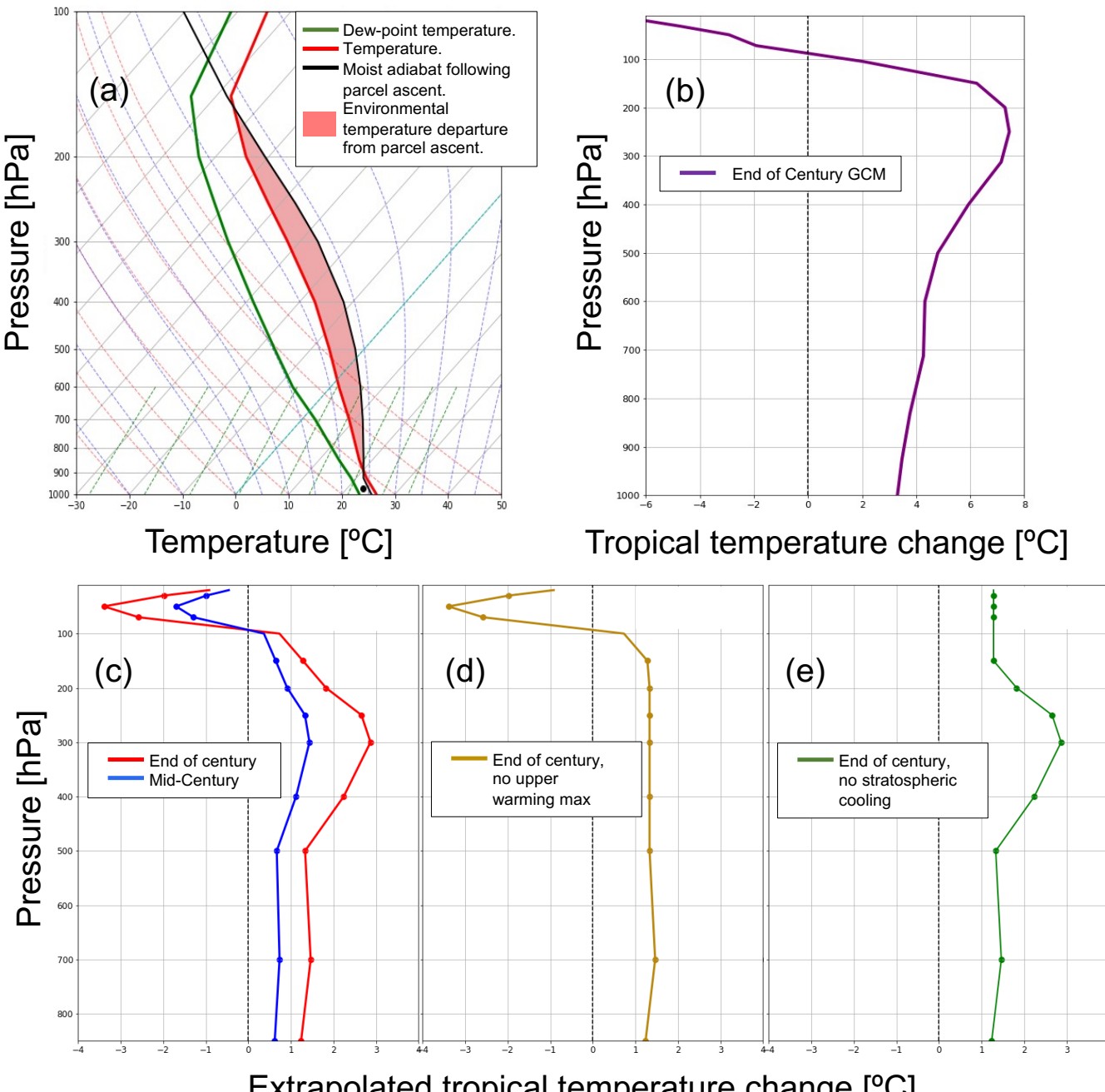

**Figure 1: (a) Dunion (2011) Moist Tropical sounding; (b) Tropical temperature change profile derived from an average of 21 CMIP5**
**GCMs under the RCP8.5 emission scenario; (c) Temperature change profiles extrapolated from hurricane-season tropical trends in**
**the RAOBCORE database and modified (d) by removal of the upper warming maximum and (e) by removal of stratospheric cooling.**
**Note the differences in vertical axis ranges between panel b and panels c,d, and e.**

# 3 Results

## 3.1 Historical temperature and tropical cyclone observations

To begin exploring whether observed changes in near-surface temperature and upper-level stratification are sufficient to explain observed trends in the TC intensity distribution, we start with an analysis of historical data. Historical summertime tropical temperature trends are compared across RAOBCORE, ERA5, and ERA-I in Fig. 2a. The known upper tropospheric warming maximum and lower stratospheric cooling are present across all three datasets but vary significantly in magnitude and vertical structure. As expected, ERA-I and RAOBCORE trend profiles agree well with each other (since ERA-I assimilates RAOBCORE data) with peak warming located at the 300 hPa level. The ERA5 exhibits 30 % weaker peak warming than RAOBCORE and locates peak warming higher in altitude, at 175 hPa. Cooling rates in the lower stratosphere are strongest in ERA5, reportedly due to the assimilation of radiosonde data adjusted by the RICH method (Haimberger et al., 2012; Hersbach et al., 2020). Simmons et al. (2014) suggest that the weaker cooling trend in ERA-I may be related to a cold bias in the lower stratosphere which persisted through the early 2000s and then was corrected through new assimilation of radio occultation data.

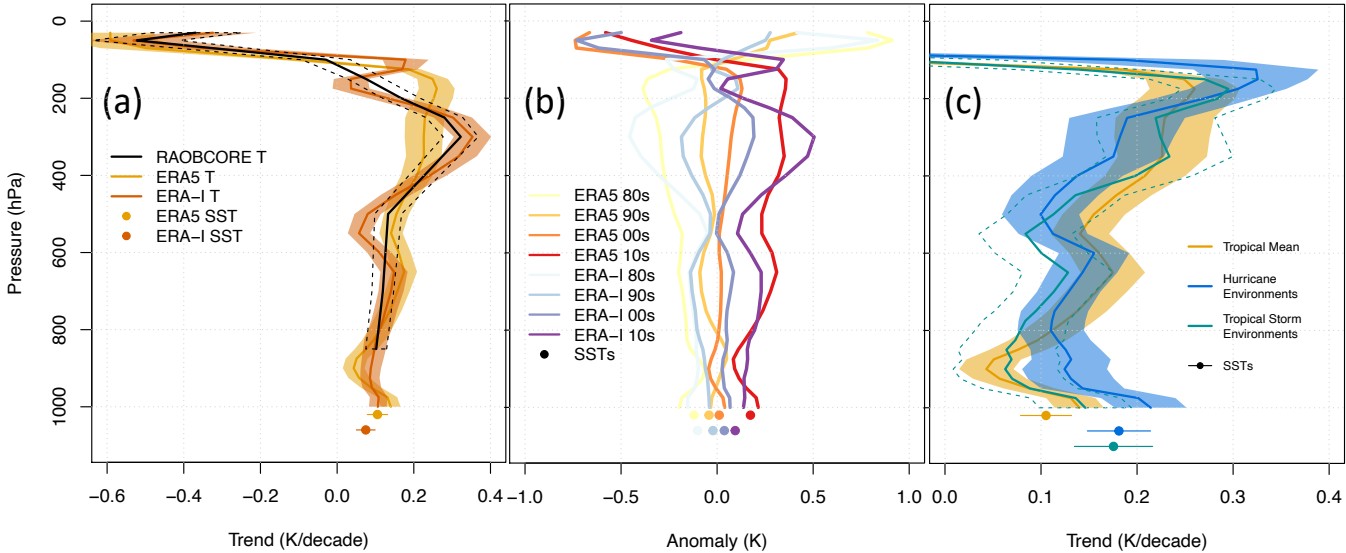

**Figure 2: Historical tropical temperature profiles averaged over 0° to 20°N for Aug-Sept-Oct and -20°S to 0° for Dec-Jan-Feb using RAOBCORE, ERA5 and ERA-I is shown as a) the linear trend over the period 1981 to 2017 (K per decade), and b) departures of decadal averages from the 1981 to 2017 average (K) for ERA5 and ERA-I only. Decadal averages are calculated over the periods 1981 to 1989, 1990 to 1999, 2000 to 2009, and 2010 to 2017. c) as in a) for ERA5 and including trends for proximal environments for tropical storms (ADT-HURSAT LMI less than 33 ms$^{-1}$) and for hurricane strength TCs (ADT-HURSAT LMI greater or equal to 33 ms$^{-1}$). Proximal environments are defined as averages within a 0.5° radius of the LMI locations two days before the TC arrives at the location using ERA5. Filled circles indicate sea surface temperatures (SSTs) where the position on the y-axis is chosen for clarity. Shading, dashed lines, and lines through the filled circles in a) and c) indicate plus/minus twice the standard error of the trend lines, approximating the 95 % confidence interval.**

314

315

We next examine whether the trend is stable across the decades, or whether the change concentrates in a particular decade. The rate of change is roughly constant across the four decades throughout the troposphere (Fig. 2b). But decadal changes in the lower stratosphere are less stable, reflecting the known step changes in temperature linked to volcanic eruptions (Ramaswamy et al., 2006).

Figure 2c shows that temperature trends proximal to strong TCs are significantly different from trends for the tropics as a whole. Proximal is defined here as an average within 0.5° of the LMI locations (according to ADT-HURSAT) two days before a TC arrives at the location. Area averaged soundings are crude approximations for the spatially varying profiles the TCs experience (e.g., Zawislak et al. 2016). However, we consider area-averaged profiles appropriate for this assessment of global trend signals, where spatial profile variations specific to individual TCs may be less important. The sample sizes are 2174 tropical storm environments and 1774 hurricane environments. Strong TC environments have warmed significantly faster than the tropical mean environment below the 850-hPa level. The SSTs in strong TC environments have also warmed faster than the tropical mean SSTs (Fig. 2c) and are likely driving the rapid warming at low levels. The warming surface and low-level temperatures would sustain the thermal disequilibrium supportive of strong potential intensities. The peak warming in the upper troposphere is correspondingly stronger for strong TC environments and located at a higher level relative to the tropics overall. Trends also differ between proximal environments for tropical storms and hurricane-strength storms, but not significantly so. Tropical storm environments also do not trend significantly differently from the tropical mean environment.

Our purpose here is not to comment on which temperature dataset produces the most accurate trends, but rather to document that the choice of temperature dataset matters for the magnitude and structure of the temperature trend. We also update previous work (Emanuel et al., 2013; Vecchi et al., 2013) that compared across reanalysis datasets by including the more recent ERA5 combined with ERA5.1. By extension, analysed relationships between TC intensity trends and near-surface temperature and upper-level stratification trends may also vary by choice of temperature dataset. Later in this section, we make links between temperature trends and TC intensity trends. This requires a temperature dataset with globally uniform coverage. We choose the ERA5 dataset for this purpose given its higher spatial resolution and newer data assimilation procedures compared to ERA-I. We next turn our attention to the changing TC intensity distribution.

At the same time as the global tropical temperatures have changed, so too has the distribution of global TC intensity. Figure 3a,b shows TC intensity distributions by historical decade in both the IBTrACS and ADT-HURSAT datasets. First, we notice the differently shaped distributions between IBTrACS and ADT-HURSDAT. Kossin et al. (2020) explain that cirrus-obscured TC eyes can cause underestimation of lifetime maximum intensity (LMI) at around 33 ms$^{-1}$. It's likely that this dataset,

therefore, over-reports LMI values less than 33 ms$^{-1}$, with higher LMI only reported if the algorithm locks onto a clearing eye
signature as TCs intensify. ADT-HURSAT, therefore, sacrifices storm-level accuracy for improved long-term statistics.

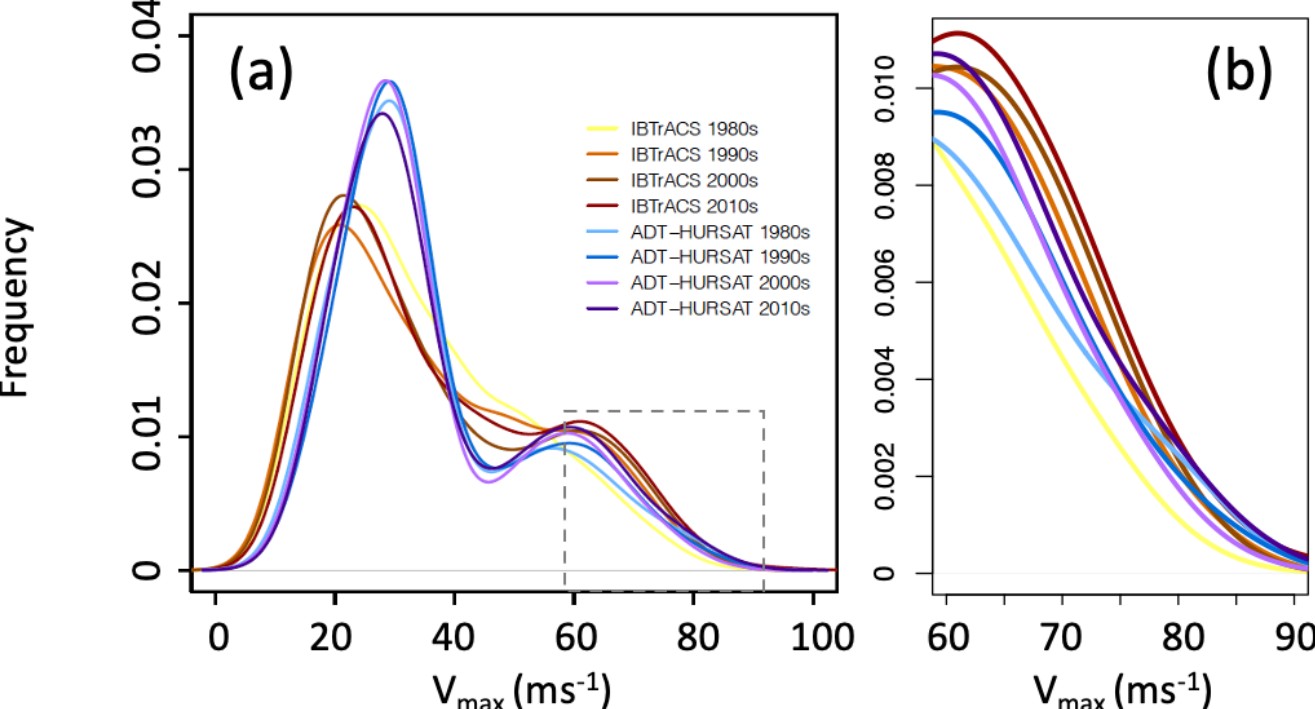


**Figure 3: a,b) Distributions of global TC LMI (lifetime maximum 1-minute sustained wind speed at 10 m above the surface, ms$^{-1}$)**
**for the period 1981 to 2017 split by historical decade using IBTrACS and ADT-HURSAT. The exact years for each decadal period**
**are 1981 to 1989, 1990 to 1999, 2000 to 2009, and 2010 to 2017. Kernel density is estimated using Gaussian smoothing kernels with**
**a standard deviation of 5 ms$^{-1}$. Panel b) provides a close-up view of the portion of panel a) outlined by the grey dashed line.**

The well-established bi-modal distribution is present in both datasets, and both reproduce the known result of an increasing
proportion of the strongest storms over time (e.g., Elsner et al., 2008; Kossin et al., 2020). We also reproduce the stronger
trends in IBTrACS than ADT-HURSAT. For the proportion of major hurricanes (category 3 and higher on the Saffir-Simpson
scale), Kossin et al. (2020) find the increase in ADT-HURSAT is about half that in IBTrACS and suggest that half the trend
in IBTrACS is attributable to changes in observing systems. When considering the proportion of category 4 and 5 storms, we
find even larger discrepancies. In IBTrACS, the proportion of category 4 and 5 storms increases from 11.3 % in the 1980s to
20.9 % in the 2010s; a factor of 1.85 increase. For ADT-HURSAT, the proportion increases from 14.1 % in the 1980s to 17.7
% in the 2010s; a factor of only 1.26, and a rate approximately 3 times lower than in IBTrACS. Our finding here is consistent
with the greater impact of observing system change for the strongest storms (Kossin et al., 2020). Interestingly, we also find
that IBTrACS produces more than half the change between the first two decades (the 1980s to the 1990s), whereas ADT-
HURSDAT produces more than half the change between the final two decades (2000s to the 2010s).

We now begin to explore statistical linkages between the changing TC intensity and near-surface and upper-level temperatures. We use quantile regression models to explore how the strength of the statistical relationship between LMI and environmental temperature varies by storm intensity, following the approach used in Elsner et al. (2008) and Kossin et al. (2013). Our quantile regression models specify how the LMI quantile changes with temperature variation. This allows us to identify whether relationships with the surface or upper-level temperature differ between strong and weak storms. We later compare these assessments to those derived from our numerical simulations.

374 We start by quantifying temporal trends in LMI to link back to existing work and provide a starting point from which to explore trends concerning temperature. When considering all TCs (Fig. 4a), only those exceeding hurricane strength (>33 ms$^{-1}$) show intensification, but trends are not significantly different from zero. Kossin et al. (2020) report that quantile regression can be highly sensitive to the range of the data. When considering only hurricane-strength storms (Fig. 4b) we found that intensification is significantly different from zero, peaking at 3 ms$^{-1}$ per decade for a hurricane quantile of 0.4. These results reproduce those of Kossin et al. (2020).

381 We next explore how these trends in LMI quantiles compare to trends in the theoretical maximum potential intensity, to determine how strong vs. weak storms have kept pace with trends in their PI. The theoretical maximum potential intensity is calculated using E-PI (Emanuel, 1988) on thermodynamic profiles from ERA5 data proximal to individual TCs at the time of LMI. The linear trend in mean E-PI is 1.2 ms$^{-1}$ per decade for locations of all TCs and 0.9 ms$^{-1}$ per decade for locations of hurricane-strength TCs only. Given that tropical storm strength TCs show no temporal trend, they have not kept pace with their rising E-PI. But hurricane-strength storms exhibit super-E-PI trends and have therefore closed the gap between realized and maximum potential intensity.

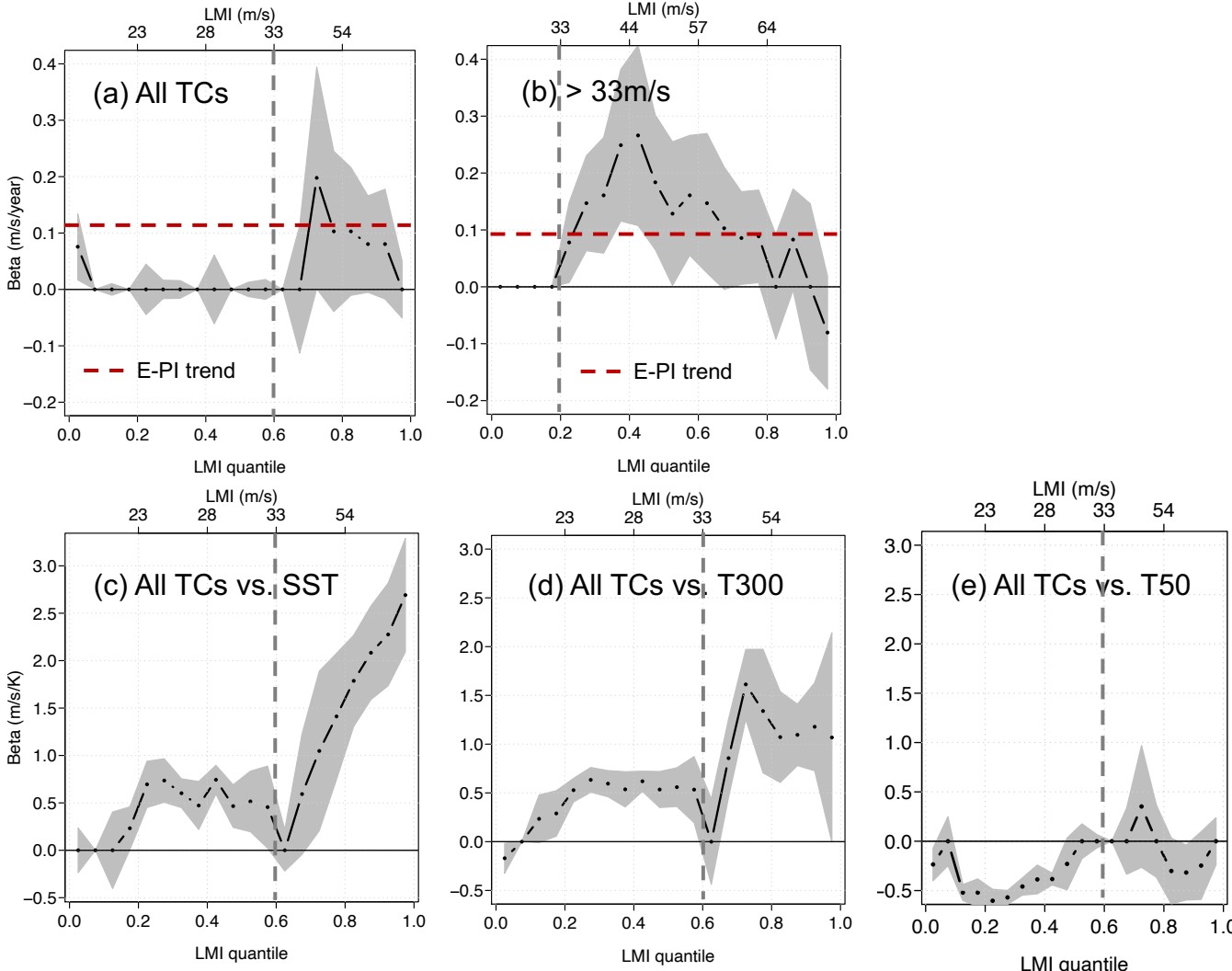

389

**Figure 4: Trends in global LMI quantiles using ADT-HURSAT over the period 1981 to 2017. a) Temporal trends for all TCs, b)**
**temporal trends for hurricane strength (>33 ms$^{-1}$) TCs only, c) trends with SST for all TCs, d) trends with temperature at the 300**
**hPa level (T300) for all TCs, and e) trends with temperature at 50 hPa (T50) for all TCs. Quantiles vary between 0.025 and 0.0975**
**with an interval of 0.05. The 95 % confidence interval (grey shading) is calculated from bootstrapping with 200 replications. The**
**grey vertical dashed lines are reference lines indicating hurricane category 1 intensity. The slope of the E-PI trend line is shown in**
**horizontal red dashed lines in a) and b). E-PI is calculated using LMI-proximal data. The second x-axis along the top of each panel**
**shows the LMI values corresponding to the LMI quantiles. In b) the second x-axis starts at 33 ms$^{-1}$ (by definition) and remains at 33**
**ms$^{-1}$ until the 0.2 quantile. R code is adapted from Elsner and Jagger (2013) available at https://rpubs.com/jelsner/5342.**


Figures 4c,d,e show relationships between LMI quantiles over all TCs and SST, temperature at the 300-hPa level (T300), and

temperature at the 50-hPa level (T50). As before for the calculation of E-PI, representative environmental temperatures are

obtained using LMI proximal values. In general, we find large and statistically significant relationships. Intensity has increased
substantially with warming SSTs almost universally across LMI quantiles, but with a markedly different response between
hurricane-strength storms and weaker storms. Tropical storm strength quantiles have increased by approximately 0.6 ms$^{-1}$ per
K, whereas the rate rises rapidly with LMI quantiles above hurricane category 1 strength, reaching a maximum of 2.6 ms$^{-1}$ per
K at the highest quantiles. This is markedly different behaviour from the temporal trends where the higher rates are located at
the middle quantiles. We also note the dip in the trend at quantiles close to about 33 ms$^{-1}$. These may not be reliable because
it coincides with the intensity at which the ADT-HURSAT determinations can be influenced by cirrus-obscured eyes.
The response of LMI quantiles to T300 is qualitatively similar to the response to SST but trends plateau for the highest
quantiles. This similarity may be expected given the strong correlation between proximal SST and proximal T300 (R = 0.78).
The reduced rates of change for the highest quantiles may also be expected given the larger change in upper tropospheric
temperature per unit change in SST. As before for SST, hurricane strength TCs exhibit markedly different behaviour to weaker
storms: They intensify with T300 warming at approximately twice the rate of weaker storms.
The response of LMI quantiles to T50 temperature (Fig. 4c) shows increasing intensity with cooling across most LMI quantiles
but is statistically significant for tropical storm strength storms only. We, therefore, do not find a significant relationship
between trends in hurricane intensity and lower stratosphere temperature, at least for this global-scale analysis. This is
consistent with the GCM study by Vecchi et al. (2013) but inconsistent with idealized simulations of Ramsay (2013).
In summary, our analysis of historical records finds that hurricane-strength storms exhibit markedly different behaviour to
weaker storms in environments of changing near-surface and upper-level temperature. Hurricane strength storm intensity
increases at twice the rate or more compared to weaker storms within environments of sea surface temperature warming.
Hurricane strength storm intensity also increases at twice the rate compared to that of weaker storms in environments of upper
tropospheric warming. Despite upper warming having a limited correlation with TC intensity, this result is perhaps
unsurprising given the strong correlation between SST and T300 (not shown). The response of hurricane-strength storms within
environments of lower stratospheric cooling was mixed and did not reach statistical significance.
**3.2 Idealized model experiments**
Towards the goal of isolating and quantifying the effects of near-surface temperature and upper-level stratification changes on
TC intensity, we turn to idealized simulations which are free from other changes. If the results of these simulations agree with
expectations, we can be more confident in attributing observed TC intensity trends to temperature changes, which are perhaps
more reliably projected by GCMs. On the other hand, if the idealized simulations indicate TC intensity trends that differ
markedly from observations, then we can be more confident that other environmental changes are dominant in driving the
observed changes. As discussed in Sect. 2.2, numerical simulations were conducted with the CM1 model in an axisymmetric
TC configuration.

The 21-member control (present climate) ensemble features an initial period of slightly weakening TC intensity, followed by
steady vortex intensification between simulation hours 12 and 90 (Fig. 5). Considerable ensemble spread develops by hour
50, with central pressure values ranging from less than 900 hPa to nearly 960 hPa at hour 100. The simulated ensemble mean
TC minimum sea-level pressure attained a minimum (maximum intensity) around hour 130, followed by slight weakening and
quasi-steady ensemble mean intensity until the end of the simulation. Simulations using a simple Newtonian cooling radiation
parameterization generally resulted in weaker TCs (blue lines in Fig. 5), motivating the use of an ensemble subset consisting
of the 13 members using more complex radiation parameterizations. The complex-radiation subset features reduced ensemble
spread, and a lower ensemble-mean central pressure (Table 2). The intensification phase of TCs in the complex radiation
members consistently begins earlier in the simulation relative to the simple-radiation subset; for instance, the time required for
Pmin to reach 960 hPa is nearly 24 hours faster for the complex radiation members (Fig. 5). We evaluate both the maximum
ensemble mean core intensity and the quasi-steady period around core intensity period later in the simulations, consistent with
"core steady-state (CS)" in the nomenclature of Rousseau-Rizzi et al. (2021). The core intensity roughly corresponds to the
LMI.

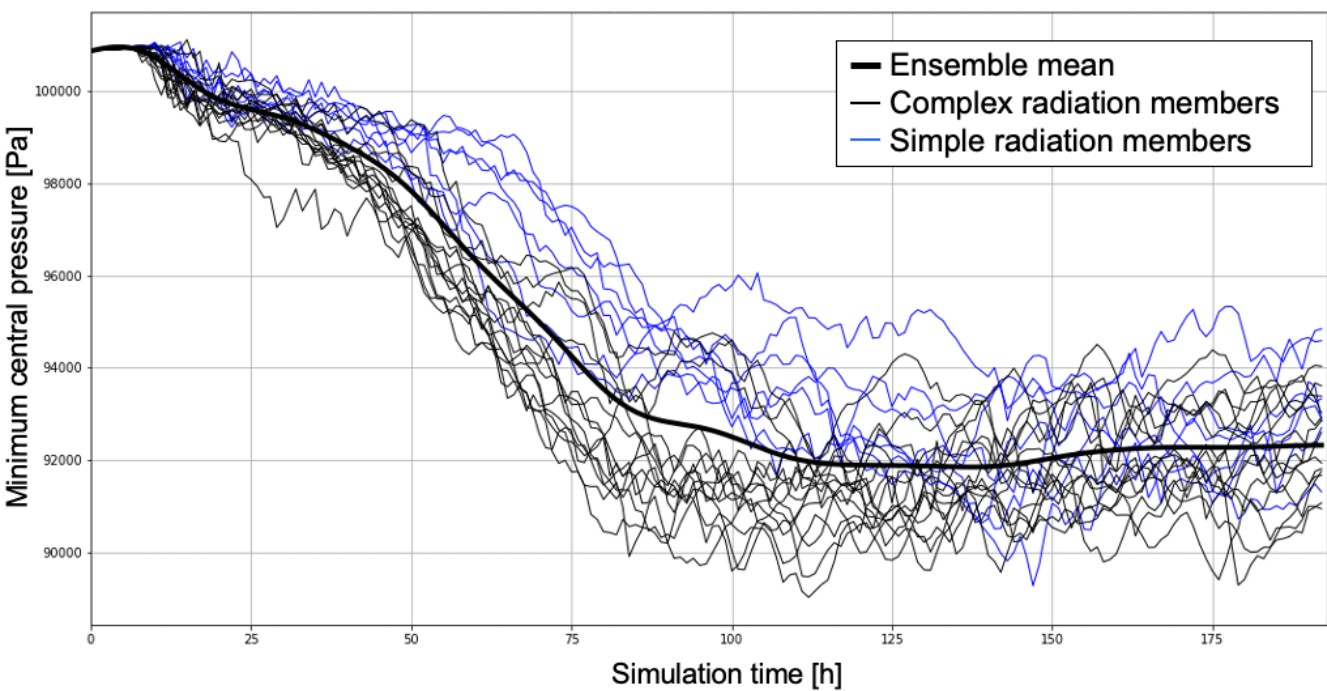


**Figure 5: CM1 time series of axisymmetric TC minimum central pressure (Pa) for the default present-day ensemble based on the**
**Dunion moist tropical sounding, distinguishing ensemble members with complex (black) and simple radiation (blue).**

For the additional experiments, time series of ensemble-mean maximum near-surface wind speed and minimum central
pressure sort out precisely as expected based on theoretical predictions: The present-day simulation features the weakest
ensemble-mean TC, while the end-of-century simulations are all stronger, with the mid-century ensemble falling between (Fig.
6, Table 2). This overall trend matches the E-PI calculations in a relative sense (Table 2). One notable difference is the removal
of the stratospheric cooling, which had no impact on E-PI but weakened the simulated storm slightly. The GCM-modified end-
of-century environment yields the greatest intensity, with filtered ensemble-mean $P_{min}$ values approaching 900 hPa in the
complex-radiation ensemble subset (Fig. 6a). This is consistent with the fact that future changes under the CMIP5 RCP8.5
scenario exceed that due to extrapolation of current observed trends (compare purple and red curves in Fig. 6a and Fig. 6b,
and abscissa values in Figs. 1b,c). In all simulations, the ensemble mean $P_{min}$ values were lower than the E-PI calculations.
Note that there is uncertainty in the E-PI calculation owing to several choices in parameter settings, as is the case with the
CM1 model. But perhaps the greatest discrepancy arises from our calculation of E-PI at the initial time, leading to possible
differences in the E-PI-calculated outflow and the realized outflow temperature in our simulations.

Each ensemble experiment exhibits considerable variability, and the ensemble standard deviations are generally larger than
the differences in the ensemble mean between the experiments (Fig. 6b, Table 2). That the relative ranking of the experimental
ensemble mean intensity matches expectation from theory is notable, but the large ensemble variability provides context
regarding statistical robustness, or lack thereof. We refrain from a dichotomous declaration of "statistically significant" or not
(e.g., Amrhein et al., 2019; Wasserstein et al., 2019). Yet, an inspection of the individual ensemble experiments demonstrates
that the relative intensity of the different ensemble members exhibits considerable consistency, motivating the use of a
Wilcoxon signed-rank test (Wilcoxon 1945), appropriate for paired samples (Fig. 6c). Except for the mid-century experiment,
small p-values relative to the present-day simulation provide more confidence in the significance of the results relative to what
comparison to the overall ensemble mean suggests (top labels in Fig. 6c). Comparison of the end-of-century with the no-upper-
warming ensemble yields a signed-rank p-value of 0.13 and compared with the no-stratospheric-cooling ensemble value of
0.29 (not shown).

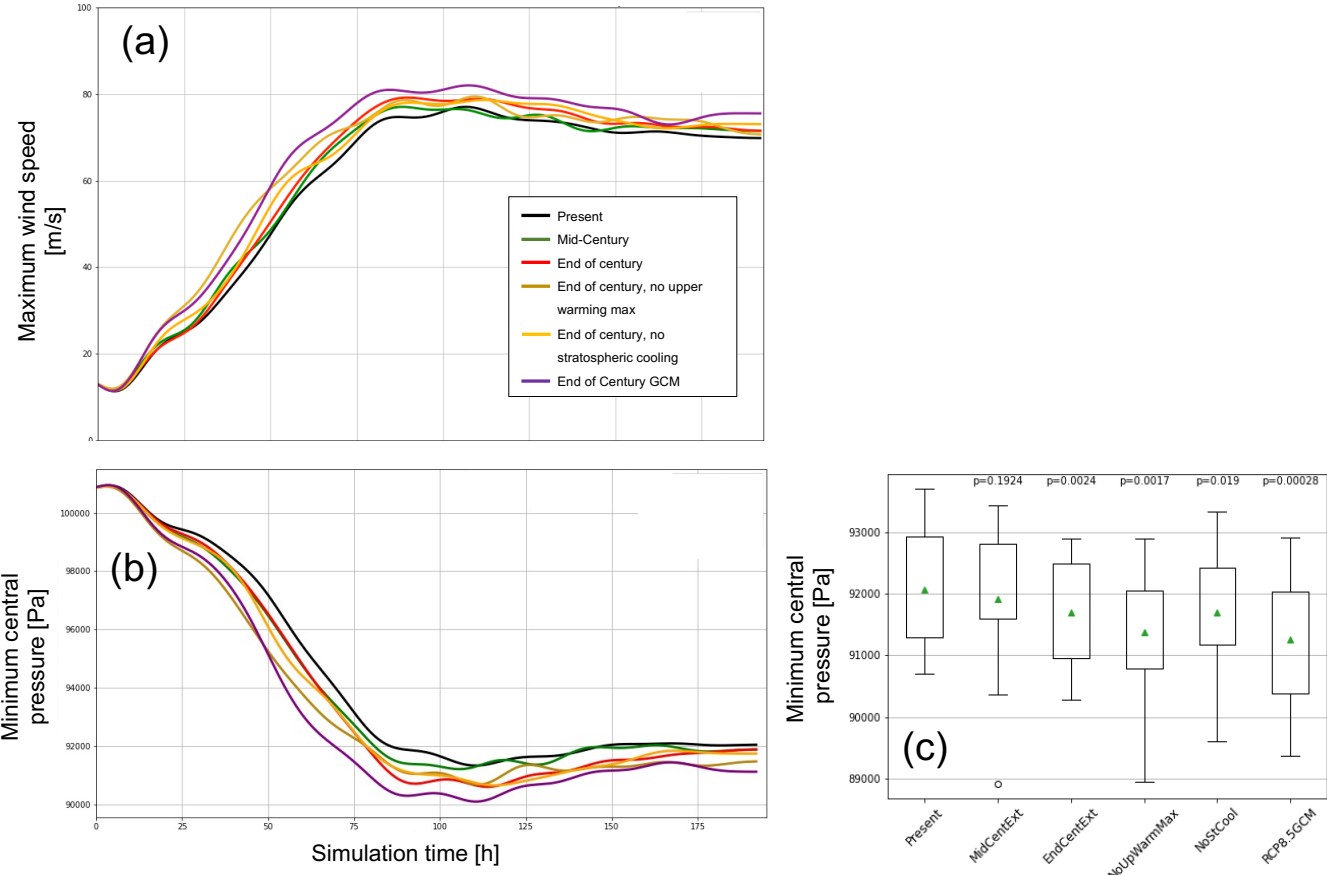


Figure 6: Time series of CM1 ensemble mean (a) maximum wind speed (ms[-1]) and (b) minimum sea level pressure (Pa) for present-day simulations with complex radiation parameterization; experiments as indicated in legend in (a). Ensemble mean time series have been smoothed with a Butterworth filter to remove high-frequency fluctuations. (c) Box plot showing the distribution of average CS period minimum central pressure over the 13 complex radiation ensemble members. Mean values are shown as green triangles, p-values from a Wilcoxon paired rank-sum test shown at the top for each experiment versus the present climate.

485

While the smoothed, ensemble mean changes are highly consistent with theoretical expectations, neither the changes predicted by E-PI theory nor those resulting from the numerical simulations are dramatic in terms of $P_{min}$. For extrapolations of current RAOBCORE trends, the end-of-century ensemble mean is characterized by $P_{min}$ values that are approximately 10 hPa lower than for the present-day ensemble. That is not to say that these intensity increases are insignificant, however. Changes in the GCM-modified environment under the RCP8.5 scenario exhibit the strongest changes in ensemble-mean $P_{min}$, approximately 12 hPa lower. The strengthening seen in the extrapolated RAOBCORE experiments is consistent with that reported for a 2 K change by Knutson et al. (2020), while the GCM experiment change, accompanied by an SST warming over 3 K, is somewhat less than what would be anticipated from the Knutson et al. (2020) review.


The consistency between the CM1 simulation results and the theoretical E-PI intensity calculations suggests that the
interpretation of the simulated TC responses to environmental change is consistent with the concept of a Carnot heat engine
(e.g., Emanuel, 1988; 1991). Because we use $P_{min}$ to measure storm intensity, we are not concerned with supergradient wind
speeds as analysed by Rousseau-Rizzi and Emanuel (2019), Hakim (2011), and Smith et al. (2008). Our hypothesis in this
analysis is that in the quiescent (un-sheared) axisymmetric CM1 environment, the TC response to changes in environmental
temperature will be consistent with PI theory and the concept of thermodynamic engines. These idealized simulations provide
an estimate of the expected effect of such changes on TC characteristics, allowing us to relate the simulation responses to the
observational TC statistics presented in Sect. 3.1.

To understand comparisons between our simulated TC intensity and E-PI changes, we compute thermodynamic efficiency and
thermodynamic disequilibrium changes in our simulations. As stated earlier, the square of PI is proportional to the product of
the thermodynamic efficiency and the thermodynamic disequilibrium (Eqn. 1 in Gilford et al. 2017). We therefore examine
whether changes in our simulated intensity ($V_{max}^2$) are proportional to simulated changes in the product of thermodynamic
efficiency and the thermodynamic disequilibrium. But first, we compare relative changes in the thermodynamic efficiency
and thermodynamic disequilibrium terms themselves.

We compute the temperature of cloudy, outflowing air in the upper troposphere for each ensemble member in each experiment,
and use this information in conjunction with SST to compute the thermodynamic efficiency (see Sect. 2.2) according to Eq.
513 (1):

$$Eff = \frac{SST - T_{\text{out}}}{T_{\text{out}}} .$$  (1)

Thermodynamic disequilibrium is computed as the difference between the saturation moist static energy at the sea surface and
a near-surface value of moist static energy. It is calculated at the initial time whereas efficiency is calculated for the CS period.

First, we examine changes in outflow temperature and pressure. The outflow temperature is remarkably similar between the
different experiments (Table 3) despite varying outflow pressures. While the warmest outflow is in the GCM-modified
experiment, as expected, this does not reach statistical significance. The similarity in outflow temperatures is consistent with
the Fixed Anvil Temperature (FAT) hypothesis (Hartmann and Larson, 2002) which argues that the environmental cooling
rate is largely governed by temperature. This follows from the saturation vapor pressure dependence on temperature via the
Clausius-Clapeyron relation. The temperature at which cooling rates rapidly decrease with height (and therefore also the
temperature of the outflow) should remain approximately constant. Surface warming, therefore, raises the altitude of the
outflow but has less effect on outflow temperature. In agreement, we find the average pressure altitude of the outflow exhibits
considerable difference among the experiments, with the present-day ensemble showing the lowest outflow altitude, and the
GCM experiment the highest (~190 hPa, Table 3). Although the differences are small relative to the ensemble standard
deviation, the no stratospheric cooling and no upper warming maximum experiments exhibit the expected changes in outflow
pressure. The FAT hypothesis could be contributing to the small changes in efficiency in our experiments with modified upper-
level stratification. Interestingly, the average outflow pressure generally reflects an altitude above the upper warming
maximum, especially for the stronger TCs in the GCM ensemble.

**Table 3: Ensemble mean thermodynamic disequilibrium, outflow temperature, outflow pressure, and thermodynamic efficiency**
**computations for the 13-member complex-radiation ensemble subset; radial wind threshold of 1.0 ms$^{-1}$ and cloud ice threshold of**
**10$^{-5}$ kg kg$^{-1}$. Ensemble standard deviation (SD) is shown for outflow temperature and pressure. Disequilibrium (defined as the**
**difference between the saturation moist static energy at the sea surface and a near-surface value of moist static energy) is calculated**
**at the initial time and all other values apply to the CS time window of the simulations, hours 150 to 192.**

| Experiment | SST (K) | Disequilibrium (J/kg) / (%) | T outflow / SD (K) | P outflow / SD (hPa) | Efficiency / % |
|---|---|---|---|---|---|
| Present-day | 301.15 | 9342.2 / -- | 224.25 / 2.73 | 216.88 / 14.89 | 0.3429 / -- |
| Mid-Century | 301.77 | 9701.0 / 3.8 | 224.22 / 3.31 | 211.92 / 17.42 | 0.3459 / 0.9 |
| End of Century | 302.39 | 10072.2 / 7.8 | 224.22 / 3.45 | 207.34 / 17.40 | 0.3486 / 1.7 |
| No upper warming max | 302.39 | 10072.2/ 7.8 | 224.08 / 3.11 | 205.87 / 15.70 | 0.3495 / 1.9 |
| No stratos. cooling | 302.39 | 10072.2/ 7.8 | 224.57 / 3.20 | 208.05 / 17.03 | 0.3465 / 1.1 |
| GCM RCP 8.5 | 304.46 | 11410.6 / 22.1 | 224.95 / 3.02 | 190.59 / 15.11 | 0.3535 / 3.1 |


For the GCM experiment, the slightly warmer outflow temperature is more than compensated by the increased SST, resulting
in the greatest thermodynamic efficiency among the experiments. The GCM experiment also produces the lowest $P_{min}$ (Table
2). The numerical simulation experiments ranked by intensity match exactly the ranking in thermodynamic efficiency (Tables
2 and 3). However, differences in thermodynamic efficiency between the ensemble members are small in magnitude, and
relative changes in thermodynamic disequilibrium with increased SST are much larger. Percent changes in disequilibrium
relative to the default run are +3.8% for the mid-century run, +7.8% for the end-of-century runs (including the no upper
warming, and no stratospheric cooling runs), and +22.1% for the GCM RCP8.5 run. Upper-level changes have no impact on
disequilibrium in our modelling. Percent changes in efficiency are much less at +.9% for the mid-century run, +1.7% for the
end-of-century runs, and +3.1% for the GCM RCP8.5 run. In contrast to disequilibrium, efficiency does change a little with
upper-level changes, but changes remain small. The lack of change in efficiency is related to the nearly constant TC outflow
temperatures between our experiments.

Having established the dominance of thermodynamic disequilibrium over thermodynamic equilibrium in driving PI, we now
examine how close our simulated intensity behaviour is to theoretical expectations. Specifically, we quantify whether our
simulated intensity changes are proportional to changes in the product of thermodynamic disequilibrium and thermodynamic
equilibrium. Quantitative comparisons are challenging given the differing absolute changes, but we do so here using percent
changes (as also used in Gilford et al. 2017). Table 4 shows close agreement between percent changes in the square of the
realized intensity and percent changes in the product of efficiency and disequilibrium. This indicates that PI theory explains
much of the TC responses to changes in environmental temperature. However, there are notable discrepancies in the
experiments with changed upper-level stratification. Possible explanations for the discrepancies are discussed in the next
section.

**Table 4: Maximum intensity ($V_{max}$) and percent changes in the left-hand side ($V_{max}^2$) and right-hand side (efficiency ×**
**disequilibrium) of Equation 1 in Gilford et al. (2017) as simulated by the complex radiation ensemble experiments. All values are**
**for time-filtered time series and represent the core steady-state (CS) period except for disequilibrium which is calculated at the**
**initial time.**

| Experiment | $V_{max}$ (m/s) | $V_{max}^2$ (%) | Efficiency × Disequilibrium (%) |
|---|---|---|---|
| Present-day | 66.14 | -- | -- |
| Mid-Century | 67.59 | 4.4 | 4.7 |
| End of Century | 69.13 | 9.3 | 9.6 |
| No upper warming max | 70.79 | 14.6 | 9.9 |
| No stratos. cooling | 69.41 | 10.1 | 8.9 |
| GCM RCP 8.5 | 74.44 | 26.7 | 25.9 |


## 4 Concluding Discussion

In a quiescent environment, theory indicates that TC intensities should exhibit considerable sensitivity to changes in near-
surface temperatures and upper-level stratification (Emanuel, 1991; Kieu and Zhang, 2018; Tao et al., 2020). In this paper, we
explored whether observed environmental temperature changes are sufficient to explain observed trends in the TC intensity
distribution, to improve the understanding and interpretation of observed and emerging trends in the TC intensity distribution.
To do so we worked to isolate and quantify the response of TC intensity to observed trends in environmental temperature using
a combination of historical data analysis and idealized numerical modelling. While our choice of axisymmetric modelling
misses potentially important TC asymmetries, such models are useful tools to begin to link theory and observations.

Our historical data analysis focused on global scales spanning four decades to emphasise the scales where thermodynamic
change is large and circulation change is minimized. Tropical storm strength intensities show no temporal trend and have
therefore not kept pace with rising PI. Hurricane strength storms, however, exhibit significant temporal trends that reach super-
PI rates for some intensity quantiles. Storms at these quantiles have therefore closed the gap between realized and maximum
potential intensity. The larger trends in the more intense storms is consistent with our finding that hurricane environments have
warmed faster than the tropical mean environment. The faster warming is most apparent in the lower troposphere and is likely
driven by faster SST warming.

The differing trends in TC environments compared to the tropical mean environment has implications for climate change
studies that use "storyline" or "Pseudo Global Warming (PGW)" methods. These methods typically apply a long time-average
change from GCMs to reanalysis conditions and uses those high-resolution conditions to drive regional model simulations of
historical and future weather events (e.g., Hazeleger et al. 2015; Lackmann, 2015; Gutmann et al., 2018; Shepherd 2019). TCs
may respond differently to environmental change more representative of that taking place locally within TC environments.

In changing our frame of reference from time to temperature, we again found markedly different sensitivities between tropical
storms and hurricane-strength storms. Hurricane strength storms intensified at up to four times the rate of tropical storms per
unit increase in surface and upper tropospheric temperature. The response of storms within environments of lower stratospheric
cooling was mixed and did not reach statistical significance. However, our global scale of analysis may miss basin-specific
sensitivities arising from the differing TC outflow layer heights relative to the tropopause (Gilford et al. (2017). SST and
outflow are strongly linked when the outflow is confined to the troposphere, but there is greater potential for larger efficiency
changes when the outflow extends above the tropopause. In addition, the differing trend magnitudes among commonly used
historical temperature and TC intensity datasets challenges our ability to understand relationships using historical data alone.

We then turned to idealized modelling to further isolate, quantify, and understand the effects of near-surface temperature and
upper-level stratification change on TC intensity, and to interpret the empirical statistics. Idealised TC simulations responded
in the expected sense to various imposed changes in environmental temperatures and generally agree with TCs operating as
heat engines. We found close agreement between percent changes in the square of the realized intensity in our simulations and
percent changes in the product of efficiency and disequilibrium. This indicates that PI theory explains much of the TC
responses to changes in environmental temperature. Removing upper tropospheric warming or stratospheric cooling from the
end-of-century experiment resulted in much smaller changes in E-PI or realized intensity than between present-day and end-

of-century. The larger proportional change in thermodynamic disequilibrium compared to thermodynamic efficiency in our experiments (in agreement with Rousseau-Rizzi and Emanuel 2021) also suggests that disequilibrium, not efficiency, is responsible for the intensity increase from present-day to end-of-century in our simulations. Possible explanations for residual differences between realized intensity change and PI change include i) necessary differences in the timing of the efficiency and disequilibrium computations, ii) limitations to the model, related to axisymmetry and parameterizations, and iii) assumptions in the E-PI algorithm.

The weak influence of lower stratospheric cooling on TC intensity in our simulations and our observational analysis is consistent with the GCM study by Vecchi et al. (2013). However, axisymmetric simulations out to radiative-convective equilibrium by Ramsay (2013) showed stronger vortex intensity with stronger imposed lower stratospheric cooling rates. This was despite much of the outflow confined to the upper troposphere. We agree with Ramsay (2013) and Ferrara et al. (2017) that it is challenging to reconcile contrasting results across different models with different parameter settings and analysis procedures, and across studies using limited historical datasets.

Analysis of TC outflow found little change in the outflow temperature but a rising mean pressure outflow altitude that is located above the altitude of peak upper tropospheric warming. The near constancy of outflow temperatures limited thermodynamic efficiency changes with surface warming, and upper level temperature change mattered less than we originally thought. The FAT hypothesis appears to explain our findings well, and would limit thermodynamic efficiency change under changed upper-level stratification. Further work is needed to understand, at a process level, the extent of applicability of the FAT hypothesis for TCs. For tropical convection it has support from observational analysis (Xu et al., 2007) and convection-resolving idealized numerical simulations (Kuang and Hartmann, 2007). Some additional supporting evidence for a FAT for TCs is provided by idealized cloud-resolving modelling (Khairoutdinov and Emanuel, 2013) and by analysis of TC cloud top temperatures in ADT-HURSAT data (Kossin, 2015). However, detecting trends in TC cloud top temperatures is complicated by a poleward trend in the latitude of LMI (Kossin, 2015).

Increasing thermodynamic disequilibrium with warming may also explain the fastest temporal trends in intensity for the middle LMI quantiles. With warming, middle LMI quantile TCs are closing the gap with PI. The strongest storms, however, were already close to their PI, and weaker storms are more strongly limited by other environmental factors such as shear or dry air. Techniques to simulate weaker storms within the idealized modelling framework are needed to test this hypothesis.

The magnitude of the simulated changes, even for extrapolated trends, is relatively small compared to observed trends in TC characteristics. This suggests that environmental temperature changes contributed to some of the observed TC intensity change, but that other environmental factors dominated as the root causes, including, for example, changes in vertical wind shear, humidity, incipient disturbances, or internal asymmetries.


Extrapolated observational temperature trends resulted in weaker TC intensity trends relative to change profiles based on an
ensemble of CMIP5 GCMs under the RCP 8.5 emission scenario. Future extensions of this work could omit the GCM-based
tropical upper warming maximum or stratospheric cooling to determine whether a more substantial change results relative to
these exercises with the extrapolated observations. The use of CMIP6 trends would also be informative. Future work could
also start from a different base sounding, other than the Dunion (2011) North Atlantic moist tropical sounding. It's possible
that different magnitude sensitivities between the historical data analysis and the idealized simulations could be due, in part,
to our use of this single profile that allows all simulated storms to reach the highest observed intensities. Base soundings
representative of the observed tropical storm and hurricane-strength storm environments may yield more nuanced sensitivity
to environmental temperature change, given permitted variations in outflow altitude. Future work should also include tests
with fully 3-D TC simulations; such simulations would include the effects of potentially important internal asymmetries and
also allow examination of changes in intensification rate and timing. Finally, more comprehensive physical process studies are
needed to interpret the empirical and idealized modelling findings reported here and work towards untangling the factors
driving observed intensity changes.

**Appendix A**
**Table A1: Description of namelist settings for axisymmetric CM1 ensemble simulations.**

| member | sfcmodel | oceanmodel | isftcflx | radopt | rterm | ptype |
|--------|----------|------------|----------|--------|-------|-------|
| 1 | 1 | 1 | 1 | 0 | 1 | 5 |
| 2 | 2 | 2 | 2 | 0 | 1 | 5 |
| 3 | 2 | 1 | 1 | 0 | 1 | 5 |
| 4 | 2 | 1 | 2 | 0 | 1 | 5 |
| 5 | 3 | 2 | 2 | 0 | 1 | 5 |
| 6 | 3 | 1 | 1 | 0 | 1 | 5 |
| 7 | 3 | 1 | 2 | 0 | 1 | 5 |
| 8 | 3 | 2 | 2 | 2 | 0 | 3 |
| 9 | 4 | 1 | 1 | 0 | 1 | 5 |
| 10 | 1 | 1 | 1 | 1 | 0 | 5 |
| 11 | 2 | 2 | 2 | 1 | 0 | 5 |
| 12 | 2 | 1 | 1 | 1 | 0 | 5 |

| | | | | | | |
|---|---|---|---|---|---|---|
| 13 | 2 | 1 | 2 | 1 | 0 | 5 |
| 14 | 6 | 1 | 1 | 1 | 0 | 5 |
| 15 | 3 | 1 | 1 | 1 | 0 | 5 |
| 16 | 6 | 1 | 2 | 1 | 0 | 3 |
| 17 | 4 | 1 | 1 | 1 | 0 | 3 |
| 18 | 2 | 2 | 2 | 2 | 0 | 3 |
| 19 | 6 | 1 | 1 | 2 | 0 | 3 |
| 20 | 4 | 1 | 1 | 2 | 0 | 3 |
| 21 | 1 | 1 | 1 | 1 | 0 | 5 |


**Code Availability**

The pyPI Python software package, developed by Daniel Gilford, is available from
https://zenodo.org/badge/latestdoi/247725622

**Code and Data Availability**

The ECMWF reanalysis datasets are available at (https://apps.ecmwf.int/datasets/). The results contain modified Copernicus
Climate Change Service information 2020. Neither the European Commission nor ECMWF is responsible for any use that
may be made of the Copernicus information or data it contains. IBTrACS data are available from NOAA
(https://www.ncdc.noaa.gov/ibtracs/). ADT-HURSAT data are available in the supporting information of Kossin et al. (2020).
RAOBCORE data are available at https://www.univie.ac.at/theoret-met/research/raobcore/. CMIP5 model output was obtained
from the Program for Climate Model Diagnosis and Intercomparison (PCMDI). The pyPI software used for the E-PI
calculations are available from Gilford (2021). R code for the quantile regression modelling presented in Fig. 4 is available at
from Elsner and Jagger (2013). The CM1 axisymmetric TC model is available from
https://www2.mmm.ucar.edu/people/bryan/cm1/

**Author Contribution**

JMD, GML, and AFP designed the analysis and experiments, and carried them out. JMD and GML prepared the manuscript
with contributions from AFP.

**Competing interests**

The authors declare that they have no conflict of interest.

**Acknowledgements**

JMD was supported by the Willis Research Network. GML was supported by National Science Foundation (NSF) grant AGS-1546743, awarded to North Carolina State University, and by the NCAR/MMM Visitor Program. We would like to acknowledge data support and high-performance computing support from Cheyenne (doi:10.5065/D6RX99HX) provided by NCAR's Computational and Information Systems Laboratory, sponsored by the National Science Foundation. This material is based upon work supported by the National Center for Atmospheric Research (NCAR); NCAR is a major facility sponsored by the National Science Foundation (NSF) under Cooperative Agreement 1852977. Raphaël Rousseau-Rizzi and an anonymous reviewer provided exceptionally constructive reviews of the initial version of this manuscript. We are grateful to NCAR's George Bryan for developing and maintaining the CM1 model, and Daniel Gilford for the pyPI software used for the E-PI calculations presented in Table 2. We thank NCAR's Chris Davis for suggestions that improved the manuscript.

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
