# Peer review of "The Response of Tropical Cyclone Intensity to Changes in"

_Weather and Climate Dynamics, 2021_

## Referee Comment (RC1)

Review of "The Response of Tropical Cyclone Intensity to Temperature Profile Change" by J.M. Done. G.M. Lackmann, and A.F. Prein

**Summary:** This study uses historical data and idealized modeling to understand how tropical cyclone (TC) intensity changes with varying profiles of atmospheric warming. While historical temperature changes coincide with stronger TC intensities, stronger TCs generally have a greater sensitivity to atmospheric profile changes than weaker TCs. Additionally, as the atmospheric profile warms, TCs interestingly tend to become more efficient heat engines. While these differences over time are robust, results from a large ensemble of simulations show that the changes due to varying the atmospheric temperature profile are small compared to observed TC intensity change trends, meaning changes to the atmospheric temperature profile cannot fully explain the observed recent increase in TC intensity.

Overall, the study was well-written and well-presented. I found two main issues which I think will be easy to address. First, the colors used in the figures, particularly Figs.1b and 2b, were incredibly hard to distinguish. Please change the colors or add symbols to the lines for them to be more distinctive. Secondly, throughout the manuscript, the authors continually refer to the results derived from axisymmetric simulations as generalized results for TC intensity. For example, at the beginning of the Concluding Discussion in L513-514, the authors state: "To do so we worked to isolate and quantify the response of TC intensity to observed trends in environmental temperature using a combination of historical data analysis and idealized numerical modelling." With very well-documented limitations of axisymmetric simulations, as well as the plethora of work about how thermodynamic and kinematic asymmetries influence TC intensity (See comment on L96-102), these generalizations about TC intensity are inappropriate. The authors do appropriately state their weaknesses in L475-480 and L573-575 to put the work in proper context, but this needs to be more clearly articulated throughout the manuscript. To be clear, I am not criticizing the choice to use axisymmetric simulations. My concern is how the significance of the results derived from the simulations are portrayed. Given that these concerns can be addressed without additional analysis, I am recommending **Acceptance after minor revisions**.

**Points by line number:**

L96-102: It is not surprising that the intensity evolution of TCs does not closely follow that of PI. There are many internal processes, particularly asymmetries in the thermodynamic (e.g., in the distribution of moist entropy, Riemer et al. 2010; Alland et al.2021a,b; Wadler et al. 2021) and dynamics (e.g., in the convective or precipitation distributions, Rogers et al. 2013; Zawislak et al. 2016; Alvey et al. 2020) which would limit the strength of a TC.

L155-156: I don't understand why the authors would not just use ERA5.1 for their entire analysis. Some clarification here would be beneficial.

L194-196: I disagree with this sentence. As alluded to in comment to L96-102, asymmetric thermodynamic processes (e.g., downdraft and radial ventilation), which occur as a response to

TC-environment interactions, would certainly "vary substantially in direct response to changes in the environmental temperature profile". This is a more significant limitation than the authors mention. Now, I do not believe this invalidates the study, but it should be emphasized further.

L224: Why did the authors change PI to E-PI while still citing the Emanuel (1988) paper for both. Some clarification is needed here.

Figure 1b: It is very difficult to distinguish the line colors, even when zoomed in 200%! Please use a more divergent colorbar. Also, for all these simulations, it is unclear how humidity was treated? Was the specific humidity or relative humidity maintained constant from the Dunion moist-tropical sounding? This would be helpful to know.

Figure 2 caption: Missing a space between "to2017" (L286)

Figure2b: Same comment in color as in Figure1b. I can't distinguish the color between ERA5 00s and ERA5 90s.

L304-306: In my experience, soundings around TCs vary substantially with radius, storm-relative location, etc (see Zawislak et al. 2016 as an example). I understand this is the best methodology for climate studies, but I wonder how representative these soundings can be of TC environments? Some comments about this in the text would be nice.

L418-419: It doesn't look to me like the stead-state intensity of the simple radiation members is that much weaker than the ensemble mean. Certainly, those members intensify at a slower rate, but by hour 150, it looks like a number of those members are stronger than the ensemble mean.

L486-488: Very small detail, but these sentences can be combined to read: "The similarity in outflow temperatures is consistent with the Fixed Anvil Temperature (FAT) hypothesis (Hartmann and Larson, 2002) which argues that the environmental cooling rate is largely governed by temperature."

L587-589: Thank you for including this contingency. While many use the Dunion moist tropical sounding, it is not realistic in many TC environments. The older Jordan mean sounding (Jordan 1958) can produce more realistic simulations.

Alland, J. J.**,** B. H. Tang, K. L. Corbosiero, and G. H. Bryan, 2021a: Synergistic effects of midlevel dry air and vertical wind shear on tropical cyclone development. Part I: Downdraft ventilation. *J. Atmos. Sci.*, **78**, 763--782, doi: 10.1175/JAS-D-20-0054.1.

Alland, J. J.**,** B. H. Tang, K. L. Corbosiero, and G. H. Bryan, 2021b: Combined effects of midlevel dry air and vertical wind shear on tropical cyclone development. Part II: Radial ventilation. *J. Atmos. Sci.*, **78**, 783--796, doi: 10.1175/JAS-D-20-0055.1.

Alvey, G.R., E. Zipser, and J. Zawislak, 2020: How does Hurricane Edouard (2014) evolve toward symmetry before rapid intensification? A high-resolution ensemble study. *J. Atmos. Sci.,* **77**, 1329-1351, https://doi.org/10.1175/JAS-D-18-0355.1

Jordan, C.L., 1958: Mean soundings for the West Indies area. *J. Meteor.* **5**, 91-97.

Riemer, M., M. T. Montgomery, and M. E. Nicholls, 2010: A new paradigm for intensity modification of tropical cyclones: Thermodynamic impact of vertical wind shear on the inflow layer. *Atmos. Chem. Phys.*, **10**, 3163–3188, doi:10.5194/acp-10-3163-2010.

Rogers, R. F., P. D. Reasor, and S. Lorsolo, 2013: Airborne Doppler observations of the inner-core structural differences between intensifying and steady-state tropical cyclones. *Mon. Wea. Rev.*, **141**, 2970–2991, doi:10.1175/MWR-D-12-00357.1.

Wadler, J.B., J.A. Zhang, B. Jaimes, and L.K. Shay, 2021: The Rapid Intensification of Hurricane Michael (2018): Storm Structure and the Relationship to Environmental and Air-Sea Interactions. *Mon. Wea. Rev.,* **149**, 1517-1534, https://doi.org/10.1175/MWR-D-20-0324.1

Zawislak, J., H. Jiang, G. R. Alvey III, E. J. Zipser, R. F. Rogers, J. A. Zhang, and S. N. Stevenson, 2016: Observations of the structure and evolution of Hurricane Edouard (2014) during intensity change. Part I: Relationship between the thermodynamic structure and precipitation. *Mon. Wea. Rev.*, **144**, 3333–3354, https://doi.org/10.1175/MWR-D-16-0018.1.

---

## Referee Comment (RC2)

**Summary**

In "The Response of Tropical Cyclone Intensity to Temperature Profile Change", Done et al evaluate how tropical cyclone intensity responds to changes in vertical temperature profile in idealized simulations. The changes in temperature profile are derived from the RAOBCORE data and from ERA5 reanalysis and are used as an initial condition to CM1 axisymmetric simulations. Resulting changes in TC intensity are then explained by invoking the concept of potential intensity, and it is found that the sign of intensity changes corresponds to the sign of the change in thermodynamic efficiency, one of the components of potential intensity. As the climate warms, TCs become more intense.

I think the paper was quite interesting to read, and that its goal represents an important endeavour in bridging the gap between theory and observations. However, there are issues in the comparison between the modelling results and the theory that need to be addressed before it can be considered for publication. For these reasons, I recommend major revisions.

**Major comments**

**1. Comparison to PI theory**

**1.1. Efficiency vs disequilibrium**

Throughout this paper, changes in intensity are discussed in the context of changes in E-PI qualitatively only via the changes in thermodynamic efficiency, which is problematic. The square of E-PI is proportional to the product of the thermodynamic efficiency and the thermodynamic disequilibrium, not only to the thermodynamic efficiency. While the role of the thermodynamic disequilibrium is acknowledged in the introduction, it is not discussed further in the paper. This is an issue because changes in thermodynamic disequilibrium, not efficiency, dominate PI variations in multiple contexts, from seasonal variations [Gilford et al., 2017] to interannual and decadal variations [Rousseau-Rizzi and Emanuel, 2021].

Table 2 clearly shows substantial variations of SST between experiments, which reinforce the idea that changes in thermodynamic disequilibrium might be dominant here too. Further, removing upper tropospheric warming or stratospheric cooling from the end-of-century experiment results in much smaller changes in E-PI or in TC minimum pressure than between present-day and end-of-century, which suggests that disequilibrium, not efficiency, is responsible for the intensity increase from present-day to end-of-century. In other words, changes in saturation enthalpy at sea-surface temperature relative to near-surface air enthalpy may be more important to changes in TC intensity than the changes in efficiency are. Hence, I think that, to interpret the results of the TC simulations presented in this study by comparing them to E-PI, it is necessary to address how disequilibrium changes by comparison to efficiency.

As a possibly useful aside: the E-PI algorithm used here is formulated following a CAPE-based definition of E-PI, which does not depend explicitly on efficiency and disequilibrium like the "Carnot" form of E-PI does. Rousseau-Rizzi et al. [2022] provides a comparison of the CAPE and Carnot forms of PI which may be useful here to assess why E-PI is changing.

**1.2. Temperature profile changes**

This study generally interprets changes in TC intensity due to changes in temperature profile in light of E-PI theory. I think this calls for a small but important clarification. Since E-PI depends on the difference between environmental CAPE and saturation CAPE at SST, most of the effect of the environmental profile on E-PI simply cancels out [e.g., Garner, 2015, Rousseau-Rizzi et al., 2022]. The only locations where the environmental profile matters for E-PI in the algorithm (unless the CIN is larger than the CAPE), are near the surface and near the storm outflow. In other words, in most cases, perturbing the temperature profile in the mid-troposphere will result in zero change in E-PI.

Hence, I think it would be useful to be just a bit more precise throughout the paper. Instead of talking about changes to the "temperature profile" at large influencing E-PI, it could be better to talk about changes in "outflow layer temperature" or "upper levels stratification" or something similar. I do not suggest that changes in mid-level temperature would have no effect on simulated TC intensity itself, but simply that any response in simulated TC intensity would be inconsistent with PI theory (which is, in itself, interesting).

**2. Numerical domain size**

At L199, it is stated that the radial domain length is 768 km which, I strongly believe, is insufficient. This is particurlarly true with cloud-radiation interactions, which you have (I think), and which tends to greatly increase the radial extent of TCs [Bu et al., 2014]. I think a radial extent of 3000 km is closer to what you might need to completely avoid having the secondary circulation impede on the outer boundary, which would influence the solution. I would have no problem with a stretched horizontal grid to help you avoid increasing your current computing cost too much if that is an issue. Hence, I think it is important to verify whether the domain is large enough, and if not, to expand it.

**Minor comments**

**Comment 1.**

L92-L94: These seem to be presented as two distinct effects, but my impression is that, perhaps, the question is whether the TC outflow temperature occurs in the lower stratosphere or in the upper troposphere. If the former, thermodynamic efficiency increases, and if the latter, it decreases, which might explain the bimodal changes in intensity with warming.

**Comment 2.**

L97-L100: This disparity between PI changes and simulated changes could be due to the outflow temperature as computed in a PI algorithm being lower than the actual outflow temperature of the storm, so that SSTs decrease in a region that could in theory impact TC intensity, but in practice does not. This is just a thought, I am not very familiar with the paper by Vecchi et al. (2013).

**Comment 3.**

Table 2: It would be nice to clarify how the variations in SST between experiments are obtained.

**Comment 4.**

L181-L186: This is an interesting idea but an odd formulation. In essence, I feel like you mean that changes in the temperature stratification near the tropopause may enhance or decrease the sensitivity of the outflow temperature to the other parameters controlling the intensity of the TC. For example $dT_{out}/dSST$ may increase if stratification decreases. Is that correct?

**Comment 5.**

L209-L210: Does lowering 1000hPa down to 1015 hPa make much of a difference? Could it be simpler to say "an sst of 28 C, close to the near-surface air temperature" if that is the case?

**Comment 6.**

L216-L217: Here and elsewhere in the paper, I think you might have switched up steady-state definitions. Did you mean "we focus on core steady-state rather than on the equilibrium state." ? In Rousseau-Rizzi et al. [2021], equilibrium state refers to periods occurring tens of days after peak intensity, while core steady-state refers to time surrounding or immediately following peak intensity (100hrs to 200hrs would fall in the core steady-state category).

**Comment 7.**

Figure 2: I think it would be very useful, and important to the interpretation, to have the corresponding SST changes and trends plotted at the bottom of each of these panels.

**Comment 8.**

L302-L303: The faster warming of strong TC environments may be due to increase in the SST heterogeneity in the tropics, leading to PI increases that are larger than the average of the tropics. The fact that this is occurring mostly under 850 hPa may be due to radiation by gravity waves homogenizing temperature above that level as explained by the WTG approximation. I think it would again be interesting to have SST changes for the corresponding profiles on plots, as I would expect that the enhanced increase in sub-850 hPa temperature is due to an locally enhanced increase in SST. If SST was not increasing faster than the tropical average in strong TC environments, the association with such a profile would lead to a decrease in thermodynamic disequilibrium and hence in PI, and probably weaker TCs.

**Comment 9.**

Figure 4b: Based on L362, I would expect the red-dashed E-PI trend to be at 0.09 m/s/year here, not 0.12 m/s/year at is appears to be

**Comment 10.**

Figure 4: I think it could be interesting to see quantile changes vs E-PI, but I leave it up to the authors to decide if they think it adds to the paper and if they want to add it.

**Comment 11.**

L433-L434: It is worth specifying in the methods whether PI in the simulations is computed a priori (e.g., using pyPI at the initial time) or in situ (e.g., in Bryan and Rotunno 2009). If, as I seem to understand, the computation of PI occurs at the initial time based on the temperature profile, this interesting discrepancy may be due to a mismatch between the E-PI-calculated outflow temperature and the realized outflow temperature of the storm.

**Comment 12.**

L441-L453: I don't know whether saying that the differences between the experiments are "small" is very useful, given the fact that the ensembles perturb numerical and physical parameters to which the TC intensity is highly sensitive. Since it is physics that is perturbed in the ensemble, not initial conditions, and we don't really expect physics to change with climate, wouldn't it make sense to simply take the central pressure difference between each corresponding ensemble member in different warming scenarios (i.e. same physics in present day and end of century)? Then the distribution of pressure differences could be used to produce a box plot of central pressure change and to verify whether change is statistically different from zero. I think this may yield similar results to the Wilcoxon test you are performing here, so please only add this if you feel that it would improve the paper. Disregard otherwise.

**Comment 13.**

L505-L508 and elsewhere: This is one instance of Major comment 1.1. For an imaginary completely fixed temperature profile, if SST increases, thermodynamic efficiency will also increase, but the biggest contribution to PI by far will come from thermodynamic disequilibrium, not efficiency. Hence this interpretation that changes in intensity are explained by the changes in efficiency is incomplete.

**\* Final note:** I really appreciated reading this paper and I hope it can be published! I cite a few papers in this revision, including some of my own, which was meant to provide examples. Please do not feel the need to cite them.
**Raphaël Rousseau-Rizzi**

**References**

Y. P. Bu, R. G. Fovell, and K. L. Corbosiero. Influence of cloud–radiative forcing on tropical cyclone structure. *Journal of the Atmospheric Sciences*, 71(5):1644–1662, 2014.

S. Garner. The relationship between hurricane potential intensity and cape. *Journal of the Atmospheric Sciences*, 72(1):141–163, 2015.

D. M. Gilford, S. Solomon, and K. A. Emanuel. On the seasonal cycles of tropical cyclone potential intensity. *Journal of Climate*, 30(16):6085–6096, 2017.

R. Rousseau-Rizzi and K. Emanuel. A weak temperature gradient framework to quantify the causes of potential intensity variability in the tropics. *Journal of Climate*, 34(21): 8669–8682, 2021.

R. Rousseau-Rizzi, R. Rotunno, and G. Bryan. A thermodynamic perspective on steady-state tropical cyclones. *Journal of the Atmospheric Sciences*, 78(2):583–593, 2021.

R. Rousseau-Rizzi, T. M. Merlis, and N. Jeevanjee. The connection between carnot and cape formulations of tc potential intensity. *Journal of Climate*, 35(3):941–954, 2022.

---

## Author Comment (AC1)

Dear Reviewer One

Thank you for your careful and extremely valuable review. You make many important points that will strengthen and help clarify our work.

This note serves as our public response to the main points raised in your review so that you and those following this discussion may get our take on the comments. A more formal point-by-point response will be forthcoming. We respond here to your main comments concerning the limitations of axisymmetric modeling, improvement of figures, and the treatment of humidity. We are also working to address all comments and incorporate all suggestions.

1) Limitations of axisymmetric modeling

Thank you for the note of caution concerning the applicability of axisymmetric model results. We agree that we over-generalized and over-extended the applicability of axisymmetric modeling throughout the paper. In our revised manuscript we shall take care to state the limitations of axisymmetric models, including stating the missed processes, and use this to interpret our results more accurately.

2) Clarity of figures

Your comment about the lack of clarity in our figures is well made. We are changing the colors to better distinguish the line plots and to adhere to color blindness standards. In looking further at Fig. 1b, we believe that the lack of clarity is due to more than just color contrasts. We will split this figure into separate panels to show the different temperature profiles more clearly.

3) Treatment of Humidity

We agree that our treatment of humidity was not clearly stated. We keep relative humidity (RH) constant, which results in increased water vapor content with warming. Some justification for this assumption is based on observations of RH to be nearly constant over the ocean over the range of seasonal temperature variation (e.g., Dai 2006; Willett et al. 2007). If RH is maintained over that large temperature range, then it is reasonable to assume that it would also be maintained over much smaller temperature increments.

Dai, A.: Recent climatology, variability, and trends in global surface humidity, J. Climate, 19, 2589–3606, https://doi.org/10.1175/JCLI3816.1, 2006.

Willett, K. M., N. P. Gillett, P. D. Jones, and P. W. Thorne, 2007: Attribution of observed surface humidity changes to human influence. Nature, 449, 710–712.

---

## Author Comment (AC2)

Dear Dr. Rousseau-Rizzi

We consider ourselves fortunate to receive this exceptionally helpful review. Your deep expertise has provided key insights that will strengthen our paper and clarify our findings.

This note serves as our public response to the major points raised in your review so that you and those following this discussion may get our take on the comments. A more formal point-by-point response will be forthcoming. We respond here to the main comments concerning the connection to PI theory and the numerical domain size. We are also working to address all comments and incorporate all suggestions.

1) Comparison to PI Theory

Thank you for alerting us to thermodynamic disequilibrium as a missed and potentially important piece connecting our modeling and theory. We agree that our substantial SST changes could affect disequilibrium. However, our corresponding changes to low level temperature may lessen any changes. We are computing the thermodynamic disequilibrium values across our simulations to explore this line of inquiry.

The suggested papers are highly relevant. After reading these papers we have a better understanding of why storm intensity peaks, then decays in the axisymmetric simulations (due to artificial drying in the subsiding branch). In our modeling, we suggest that this is less of an issue since we are studying the earlier peak in intensity (which we shall be sure to state correctly). But we shall certainly refer to this to help interpretation. Thank you also for pointing out the Carnot vs. CAPE-based forms of E-PI. This is something we had not fully appreciated before receiving your review.

Your comment about E-PI being most sensitive to temperatures near the surface and at the outflow level is well made. We shall certainty emphasize these levels in a revised interpretation. In addition, we shall add some discussion of the basin-dependency of seasonal variations in the outflow reaching the lower stratosphere.

2) Numerical domain size

We are sorry for incorrectly stating our domain size to be 768 km in the original submission. We had overlooked the fact that we used a stretched grid, where an outer portion of the domain (at radial distances greater than 280 km) stretches to larger grid spacing. The domain size reported in the paper is therefore incorrect: It is 1500 km in the radial direction. We will of course update this in the revised paper.

We therefore suggest domain size is a less serious issue than it first appeared. To demonstrate this, we are making domain size sensitivity runs to confirm that the sensitivity is small compared to changes in physics options or responses to temperature

profile change. These larger domain runs double the radial distance to 3000 km and also double the radial distance beyond which stretching begins (to 560 km).

---

## Author Response (AR1)

We thank Reviewer #1 one for their careful review of our work. Our responses to the reviewer's main comments about the limitations of axisymmetric modeling, improvement of figures, and the treatment of humidity have strengthened and clarified our paper. Our specific point-by-point responses are described in blue below.

Summary:
This study uses historical data and idealized modeling to understand how tropical cyclone (TC) intensity changes with varying profiles of atmospheric warming. While historical temperature changes coincide with stronger TC intensities, stronger TCs generally have a greater sensitivity to atmospheric profile changes than weaker TCs. Additionally, as the atmospheric profile warms, TCs interestingly tend to become more efficient heat engines. While these differences over time are robust, results from a large ensemble of simulations show that the changes due to varying the atmospheric temperature profile are small compared to observed TC intensity change trends, meaning changes to the atmospheric temperature profile cannot fully explain the observed recent increase in TC intensity.

Overall, the study was well-written and well-presented. I found two main issues which I think will be easy to address.

First, the colors used in the figures, particularly Figs.1b and 2b, were incredibly hard to distinguish. Please change the colors or add symbols to the lines for them to be more distinctive.

We agree about the lack of clarity in some of our figures. We changed the colors in Fig 1 and Fig 2b to better distinguish the line plots and to adhere to color blindness standards. In looking further at Fig. 1b, we believe that the lack of clarity is due to more than just color contrasts. We therefore also split this figure into separate panels to show the different temperature profiles more clearly.

Secondly, throughout the manuscript, the authors continually refer to the results derived from axisymmetric simulations as generalized results for TC intensity. For example, at the beginning of the Concluding Discussion in L513-514, the authors state: "To do so we worked to isolate and quantify the response of TC intensity to observed trends in environmental temperature using a combination of historical data analysis and idealized numerical modelling." With very well-documented limitations of axisymmetric simulations, as well the plethora of work about how thermodynamic and kinematic asymmetries influence TC intensity (See comment on L96-102), these generalizations about TC intensity are inappropriate. The authors do appropriately state their weaknesses in L475-480 and L573-575 to put the work in proper context, but this needs to be more clearly articulated throughout the manuscript. To be clear, I am not criticizing the choice to use axisymmetric simulations. My concern is how the significance of the results derived from the simulations are portrayed. Given that these concerns can be addressed without additional analysis, I am recommending Acceptance after minor revisions.

Thank you for the note of caution concerning the applicability of axisymmetric model results. We agree that we over-generalized and over-extended the applicability of axisymmetric modeling throughout the paper. In our revised manuscript now state the limitations of axisymmetric models, including stating the missed processes, and use this to interpret our results more accurately. Our specific changes are explained below in response to your specific comments on this topic.

Points by line number:
L96-102: It is not surprising that the intensity evolution of TCs does not closely follow that of PI. There are many internal processes, particularly asymmetries in the thermodynamic (e.g., in the distribution of moist entropy, Riemer et al. 2010; Alland et al.2021a,b; Wadler et al. 2021) and dynamics (e.g., in the convective or precipitation distributions, Rogers et al. 2013; Zawislak et al. 2016; Alvey et al. 2020) which would limit the strength of a TC.

Thank you for alerting us to these recent papers showing the importance of asymmetries in TC thermodynamics and kinematics. In the revised paper we highlight the potential importance of these missed processes in the introduction, methods, and conclusions (and now cite the suggested papers), and we more carefully interpret our idealized modeling results.

L155-156: I don't understand why the authors would not just use ERA5.1 for their entire analysis. Some clarification here would be beneficial.

ERA5.1 is a rerun of ERA5 for the period 2000-2006 only. We added this statement to the revised manuscript.

L194-196: I disagree with this sentence. As alluded to in comment to L96-102, asymmetric thermodynamic processes (e.g., downdraft and radial ventilation), which occur as a response to TC-environment interactions, would certainly "vary substantially in direct response to changes in the environmental temperature profile". This is a more significant limitation than the authors mention. Now, I do not believe this invalidates the study, but it should be emphasized further.

Thank you for alerting us to our unsubstantiated assertion that there are no reasons to think 3-d effects would be important. You have provided many references that show good reason to think so. We have removed this sentence from the revised version and better highlight the limitation of 2-d modeling.

L224: Why did the authors change PI to E-PI while still citing the Emanuel (1988) paper for both. Some clarification is needed here.

Thank you. We are now more careful with these terms. The revised manuscript uses 'E-PI' to refer to the specific algorithm we used to compute potential intensity. We use 'PI' to refer to the theory of potential intensity more generally.

Figure 1b: It is very difficult to distinguish the line colors, even when zoomed in 200%! Please use a more divergent colorbar. Also, for all these simulations, it is unclear how humidity was treated? Was the specific humidity or relative humidity maintained constant from the Dunion moist-tropical sounding? This would be helpful to know.

As indicated earlier, we have redrawn Fig 1b and split the profiles over 3 separate panel plots.

We agree that our treatment of humidity was not clearly stated. We keep relative humidity (RH) constant, which results in increased water vapor content with warming. Some justification for this assumption is based on observations of RH to be nearly constant over the ocean over the range of seasonal temperature variation (e.g., Dai 2006; Willett et al. 2007). If RH is maintained over that large temperature range, then it is reasonable to assume that it would also be maintained over much smaller temperature increments.

Dai, A.: Recent climatology, variability, and trends in global surface humidity, J. Climate, 19, 2589–3606, https://doi.org/10.1175/JCLI3816.1, 2006.

Willett, K. M., N. P. Gillett, P. D. Jones, and P. W. Thorne, 2007: Attribution of observed surface humidity changes to human influence. Nature, 449, 710–712.

Figure 2 caption: Missing a space between "to2017" (L286)

Corrected. Thank you.

Figure2b: Same comment in color as in Figure1b. I can't distinguish the color between ERA5 00s and ERA5 90s.

As indicated earlier, Fig 2b has been redrawn using greater discrimination between colors.

L304-306: In my experience, soundings around TCs vary substantially with radius, stormrelative location, etc (see Zawislak et al. 2016 as an example). I understand this is the best methodology for climate studies, but I wonder how representative these soundings can be of TC environments? Some comments about this in the text would be nice.

Thank you for making us think more carefully about the representativeness of an area-averaged sounding. We agree about the potential lack of representativeness of the area

average for individual TCs. We view area averaged soundings as crude approximations for the profiles the TCs experience. We consider them appropriate for the assessment of global trend signals, where TC-specific spatial profile variations may be less important. We now state this in the revised manuscript.

L418-419: It doesn't look to me like the stead-state intensity of the simple radiation members is that much weaker than the ensemble mean. Certainly, those members intensify at a slower rate, but by hour 150, it looks like a number of those members are stronger than the ensemble mean.

This is a great point. While the choice of radiation separates the core intensity (also see Table 2 top row), there is no apparent separation of equilibrium intensity. We choose not to pursue this further, given that the core intensity corresponds more to the Lifetime Maximum Intensity in the observations and that subsequent analysis continues with only the complex radiation subset.

L486-488: Very small detail, but these sentences can be combined to read: "The similarity in outflow temperatures is consistent with the Fixed Anvil Temperature (FAT) hypothesis (Hartmann and Larson, 2002) which argues that the environmental cooling rate is largely governed by temperature."

Changed. Thank you.

L587-589: Thank you for including this contingency. While many use the Dunion moist tropical sounding, it is not realistic in many TC environments. The older Jordan mean sounding (Jordan 1958) can produce more realistic simulations.

We had originally planned to conduct our idealized model experiments over a range of TC intensities, but it proved too problematic for this study to try to control peak intensity ranges. Further efforts using different base profiles representative of different TC intensities in different basins are needed. Thank you for the note about success with the Jordan sounding.

Alland, J. J., B. H. Tang, K. L. Corbosiero, and G. H. Bryan, 2021a: Synergistic effects of midlevel dry air and vertical wind shear on tropical cyclone development. Part I: Downdraft ventilation. J. Atmos. Sci., 78, 763--782, doi: 10.1175/JAS-D-20-0054.1.

Alland, J. J., B. H. Tang, K. L. Corbosiero, and G. H. Bryan, 2021b: Combined effects of midlevel dry air and vertical wind shear on tropical cyclone development. Part II: Radial ventilation. J. Atmos. Sci., 78, 783--796, doi: 10.1175/JAS-D-20-0055.1.

Alvey, G.R., E. Zipser, and J. Zawislak, 2020: How does Hurricane Edouard (2014) evolve toward symmetry before rapid intensification? A high-resolution ensemble study. J. Atmos. Sci., 77, 1329-1351, https://doi.org/10.1175/JAS-D-18-0355.1

Jordan, C.L., 1958: Mean soundings for the West Indies area. J. Meteor. 5, 91-97.

Riemer, M., M. T. Montgomery, and M. E. Nicholls, 2010: A new paradigm for intensity modification of tropical cyclones: Thermodynamic impact of vertical wind shear on the inflow layer. Atmos. Chem. Phys., 10, 3163–3188, doi:10.5194/acp-10-3163-2010.

Rogers, R. F., P. D. Reasor, and S. Lorsolo, 2013: Airborne Doppler observations of the inner-core structural differences between intensifying and steady-state tropical cyclones. Mon. Wea. Rev., 141, 2970–2991, doi:10.1175/MWR-D-12-00357.1.

Wadler, J.B., J.A. Zhang, B. Jaimes, and L.K. Shay, 2021: The Rapid Intensification of Hurricane Michael (2018): Storm Structure and the Relationship to Environmental and Air-Sea Interactions. Mon. Wea. Rev., 149, 1517-1534, https://doi.org/10.1175/MWR-D-20-0324.1

Zawislak, J., H. Jiang, G. R. Alvey III, E. J. Zipser, R. F. Rogers, J. A. Zhang, and S. N. Stevenson, 2016: Observations of the structure and evolution of Hurricane Edouard (2014) during intensity change. Part I: Relationship between the thermodynamic structure and precipitation. Mon. Wea. Rev., 144, 3333–3354, https://doi.org/10.1175/MWR-D-16-0018.1.

Dear Dr. Rousseau-Rizzi

We consider ourselves fortunate to receive this exceptionally helpful review. Your deep expertise has provided key insights that will strengthen our paper and clarify our findings. Our responses to your main comments about comparison to PI theory, and numerical domain size have strengthened and clarified our paper. Our specific point-by-point responses are described in blue below.

**Summary**

In "The Response of Tropical Cyclone Intensity to Temperature Profile Change", Done et al evaluate how tropical cyclone intensity responds to changes in vertical temperature profile in idealized simulations. The changes in temperature profile are derived from the RAOBCORE data and from ERA5 reanalysis and are used as an initial condition to CM1 axisymmetric simulations. Resulting changes in TC intensity are then explained by invoking the concept of potential intensity, and it is found that the sign of intensity changes corresponds to the sign of the change in thermodynamic efficiency, one of the components of potential intensity. As the climate warms, TCs become more intense.

I think the paper was quite interesting to read, and that its goal represents an important endeavour in bridging the gap between theory and observations. However, there are issues in the comparison between the modelling results and the theory that need to be addressed before it can be considered for publication. For these reasons, I recommend major revisions.

**Major comments**

**1. Comparison to PI theory**
**1.1. Efficiency vs disequilibrium**

Throughout this paper, changes in intensity are discussed in the context of changes in E-PI qualitatively only via the changes in thermodynamic efficiency, which is problematic. The square of E-PI is proportional to the product of the thermodynamic efficiency and the thermodynamic disequilibrium, not only to the thermodynamic efficiency. While the role of the thermodynamic disequilibrium is acknowledged in the introduction, it is not discussed further in the paper. This is an issue because changes in thermodynamic disequilibrium, not efficiency, dominate PI variations in multiple contexts, from seasonal variations [Gilford et al., 2017] to interannual and decadal variations [Rousseau-Rizzi and Emanuel, 2021].

Table 2 clearly shows substantial variations of SST between experiments, which reinforce the idea that changes in thermodynamic disequilibrium might be dominant here too. Further, removing upper tropospheric warming or stratospheric cooling from the end-of-century experiment results in much smaller changes in E-PI or in TC minimum pressure than between present-day and end-of-century, which suggests that

disequilibrium, not efficiency, is responsible for the intensity increase from present-day to end-of-century. In other words, changes in saturation enthalpy at sea-surface temperature relative to near-surface air enthalpy may be more important to changes in TC intensity than the changes in efficiency are. Hence, I think that, to interpret the results of the TC simulations presented in this study by comparing them to E-PI, it is necessary to address how disequilibrium changes by comparison to efficiency.

As a possibly useful aside: the E-PI algorithm used here is formulated following a CAPE based definition of E-PI, which does not depend explicitly on efficiency and disequilibrium like the "Carnot" form of E-PI does. Rousseau-Rizzi et al. [2022] provides a comparison of the CAPE and Carnot forms of PI which may be useful here to assess why E-PI is changing.

Thank you for alerting us to thermodynamic disequilibrium as a missed and important piece connecting our modeling and theory. The papers you cite are highly relevant and have significant bearing on the interpretation of our results. We agree that our substantial SST changes should affect disequilibrium.

We computed disequilibrium and efficiency in our simulations and compared their relative changes. Quantitative comparisons between changes in disequilibrium and changes in efficiency are challenging, but we did so using percent changes. This removes the effect of differing absolute values and give a general sense of their relative importance (as also used in Gilford et al. 2017).

We find that percent changes in disequilibrium relative to the default run are +3.8% for the mid-century run, +7.8% for the end-of-century runs (including the no upper warming, and no stratospheric cooling runs), and +22.1% for the GCM RCP8.5 run. Upper-level changes therefore have no impact on disequilibrium in our modeling (as you anticipated). Percent changes in efficiency are much less (as you also suspected) at +.9% for the mid-century run, +1.7% for the end-of-century runs, and +3.1% for the GCM RCP8.5 run. In contrast to disequilibrium, efficiency does change a little with upper-level changes, but changes remain small. Columns showing these new computations have been added to Table 3 in the revised manuscript.

We then evaluated the PI relation (equation (1) in Gilford et al. 2017) to see how well our simulated $V_{max}^2$ changes in proportion to changes in the product of thermodynamic disequilibrium and thermodynamic equilibrium (as expected by theory). A new Table 4 (shown below) shows close agreement between percent changes in the square of the realized intensity, and percent changes in the product of efficiency and disequilibrium. This indicates that PI theory explains much of the TC responses to changes in environmental temperature. However, there are notable discrepancies for the experiments with changed upper-level stratification.

Table 4: Maximum intensity ($V_{max}$) and percent changes in the left-hand side ($V_{max}^2$) and right-hand side (efficiency × disequilibrium) of Equation 1 in Gilford et al. (2017) as simulated by the complex radiation

ensemble experiments. All values are for time-filtered time series and represent the core steady-state (CS) period except for disequilibrium which is calculated at initial time.

| Experiment | $V_{max}$ (m/s) | $V_{max}^2$ (%) | Efficiency × Disequilibrium (%) |
|---|---|---|---|
| Present-day | 66.14 | -- | -- |
| Mid-Century | 67.59 | 4.4 | 4.7 |
| End of Century | 69.13 | 9.3 | 9.6 |
| No upper warming max | 70.79 | 14.6 | 9.9 |
| No stratos. cooling | 69.41 | 10.1 | 8.9 |
| GCM RCP 8.5 | 74.44 | 26.7 | 25.9 |

There are several candidates that could explain these discrepancies (some of which you already suggested to us):

1) Differences in the timing of the computations: E-PI and disequilibrium are computed at initial time when we have most control and better able to make interpretations, whereas the efficiency computations are based on values obtained from outflowing cloudy air in the model during a certain time period, and the environment has already evolved to some extent by that time.
2) There are limitations to the model, related to axisymmetry and parameterizations. For example, there are effects in the E-PI such as the precipitation mass sink which may not be fully represented.
3) There are assumptions in the E-PI calculation, such as the wind-reduction coefficient and enthalpy-drag coefficient ratio that may lead to discrepancies.

We conclude that the simulated storms exhibit greater intensification in the differing thermodynamic settings than can be explained by a basic application of E-PI theory.  Another conclusion is that the disequilibrium change is larger than the outflow efficiency change, consistent with the papers you pointed us to. Thank you again for alerting us to our oversight.

We added a discussion of these new results. We also added a new panel plot of Vmax to Fig. 6 to better connect results with theory than just showing Pmin.

Finally, the Carnot versus CAPE versions of potential intensity is something we didn't fully appreciate before reading your paper on this. Understanding this has helped our interpretation. Thank you.

1.2. Temperature profile changes

This study generally interprets changes in TC intensity due to changes in temperature profile in light of E-PI theory. I think this calls for a small but important clarification. Since E-PI depends on the difference between environmental CAPE and saturation CAPE at SST, most of the effect of the environmental profile on E-PI simply cancels out [e.g., Garner, 2015, Rousseau-Rizzi et al., 2022]. The only locations where the environmental profile matters for E-PI in the algorithm (unless the CIN is larger than the CAPE), are near the surface and near the storm outflow. In other words, in most cases, perturbing the temperature profile in the mid-troposphere will result in zero change in E-PI.

Hence, I think it would be useful to be just a bit more precise throughout the paper. Instead of talking about changes to the "temperature profile" at large influencing E-PI, it could be better to talk about changes in "outflow layer temperature" or "upper levels stratification" or something similar. I do not suggest that changes in mid-level temperature would have no effect on simulated TC intensity itself, but simply that any response in simulated TC intensity would be inconsistent with PI theory (which is, in itself, interesting).

Your comment about E-PI being most sensitive to temperatures near the surface and at the outflow level is well made. We have emphasized these levels throughout the revised manuscript. In addition, we added some discussion of the basin-dependency of seasonal variations in the outflow reaching the lower stratosphere (Gilford et al. 2017), and its relationship with our results.

**2. Numerical domain size**

At L199, it is stated that the radial domain length is 768 km which, I strongly believe, is insufficient. This is particularly true with cloud-radiation interactions, which you have (I think), and which tends to greatly increase the radial extent of TCs [Bu et al., 2014]. I think a radial extent of 3000 km is closer to what you might need to completely avoid having the secondary circulation impede on the outer boundary, which would influence the solution. I would have no problem with a stretched horizontal grid to help you avoid increasing your current computing cost too much if that is an issue. Hence, I think it is important to verify whether the domain is large enough, and if not, to expand it.

We are sorry for incorrectly stating our domain size to be 768 km in the original submission. We had overlooked the fact that we used a stretched grid, where an outer portion of the domain (at radial distances greater than 280 km) stretches to larger grid spacing. The domain size reported in the paper is therefore incorrect: It is 1500 km in the radial direction. This has been corrected in the revised paper.

We therefore suggest domain size is a less serious issue than it first appeared. To demonstrate this, we reran our 11-member ensemble under present climate using a larger

domain with double the radial distance (3000 km) and also double the radial distance beyond which stretching begins (560 km).

Figure R1 shows the present-day ensemble minimum pressure solutions on the two domains. The final ensemble mean pressure is a little lower on the larger domain than the original domain but the difference is small relative to the ensemble spread. There also do not appear to be critical interactions between the domain size and the model physics sensitivities. Key to the validity of our results is that the sensitivity is small (particularly for the 'core' strength) compared to changes in physics options or responses to temperature profile change (see Figs. 5 and 6, in the manuscript). We note this domain size sensitivity in the revised manuscript but choose not to include these new figures.

[Figure]

*Figure R1: CM1 time series of axisymmetric TC minimum central pressure (Pa) for the present-day ensemble based on the Dunion moist tropical sounding using a 1500km radial domain size (black) and a 3000 km radial domain size (red).*

We also checked sensitivity of the maximum wind speed for a single ensemble member (Fig. R2). The two runs are nearly the same, and consistent with the minimum pressure differences. The cloud field for a single ensemble member on the original domain (Fig R3) also doesn't suggest any evidence of boundary issues.

[Figure]

*Figure R2: CM1 time series of axisymmetric TC (a) maximum wind speed (m/s) and (b) minimum central pressure (Pa) for a present-day ensemble member using the original 1500km radial domain size (blue) and a 3000 km radial domain size (orange).*

[Figure]

*Figure R3: Height-radius cross section of hydrometeors in a single present-day CM1 ensemble member suing the original domain size. Rain water and cloud water are shown in greens, and snow, graupel, and cloud ice are shown in blues and purples.*

**Minor comments**

**Comment 1**

L92-L94: These seem to be presented as two distinct effects, but my impression is that, perhaps, the question is whether the TC outflow temperature occurs in the lower stratosphere or in the upper troposphere. If the former, thermodynamic efficiency increases, and if the latter, it decreases, which might explain the bimodal changes in intensity with warming.

Yes, we agree that our presentation of two distinct effects is perhaps misleading. We have changed our description following your suggestion. Thank you.

**Comment 2.**

L97-L100: This disparity between PI changes and simulated changes could be due to the outflow temperature as computed in a PI algorithm being lower than the actual outflow temperature of the storm, so that SSTs decrease in a region that could in theory impact TC intensity, but in practice does not. This is just a thought, I am not very familiar with the paper by Vecchi et al. (2013).

Thank you for this comment. Vecchi et al. (2013) don't comment explicitly on the possible difference between PI-outflow and realized outflow. But we include a note about this as a potential contributor to disparities between PI and numerical simulations. At the suggestion of another reviewer we additionally discuss other environmental factors, and processes internal to the TC, that can affect realized intensity.

**Comment 3.**

Table 2: It would be nice to clarify how the variations in SST between experiments are

obtained.

As for the current climate SST value, SST values for the mid- and end-of-century experiments were chosen to be close to the near-surface air temperature. This is now stated in the revised manuscript.

**Comment 4.**

L181-L186: This is an interesting idea but an odd formulation. In essence, I feel like you mean that changes in the temperature stratification near the tropopause may enhance or decrease the sensitivity of the outflow temperature to the other parameters controlling the intensity of the TC. For example $dT_{out}/dSST$ may increase if stratification decreases. Is that correct?

We appreciate your framing of our hypothesis in the broader context of other parameters controlling the intensity of the TC. Thank you. We have reworded this paragraph.

**Comment 5.**

L209-L210: Does lowering 1000hPa down to 1015 hPa make much of a difference? Could it be simpler to say "an sst of 28 C, close to the near-surface air temperature" if that is the case?

We agree that our description was unnecessarily detailed. We have simplified the explanation as suggested.

**Comment 6.**

L216-L217: Here and elsewhere in the paper, I think you might have switched up steady-state definitions. Did you mean "we focus on core steady-state rather than on the equilibrium state." ? In Rousseau-Rizzi et al. [2021], equilibrium state refers to periods occurring tens of days after peak intensity, while core steady-state refers to time surrounding or immediately following peak intensity (100hrs to 200hrs would fall in the core steady-state category).

Thank you. We do indeed mean core steady-state rather than equilibrium state. We also note the peak core state, given its approximate equivalence to LMI. This has been corrected throughout. We also make note of the artificial drying leading to lower equilibrium intensity in CM1.

**Comment 7.**

Figure 2: I think it would be very useful, and important to the interpretation, to have the corresponding SST changes and trends plotted at the bottom of each of these panels.

The corresponding SST changes have been added to Fig. 2. Please see our response to your related comment below for more details.

**Comment 8.**

L302-L303: The faster warming of strong TC environments may be due to increase in the SST heterogeneity in the tropics, leading to PI increases that are larger than the average of the tropics. The fact that this is occurring mostly under 850 hPa may be due to radiation by gravity waves homogenizing temperature above that level as explained by the WTG approximation. I think it would again be interesting to have SST changes for the corresponding profiles on plots, as I would expect that the enhanced increase in sub-850 hPa temperature is due to an locally enhanced increase in SST. If SST was not increasing faster than the tropical average in strong TC environments, the association with such a profile would lead to a decrease in thermodynamic disequilibrium and hence in PI, and probably weaker TCs.

Thank you for sharing this compelling hypothesis. We added presentations of the SST data to the panel plots in Fig. 2. We see consistency between the lower-level temperatures and SST trends in strong TC environments, with both increasing faster than the tropical mean. SSTs in tropical storm environments also warm faster that the tropical mean, and faster that the overlying air temperature. We don't understand this without further investigation and choose not to expand further here.

Another intriguing detail is that SST trend magnitudes are less than lower-level air temperature trend magnitudes in all of the panel plots. Fig. 2b breaks it down by decade with SST anomalies warmer than the overlying air temperature in the early period and SSTs cooler than the overlying temperature in the more recent period. We choose not to dwell on this detail in the manuscript, but it certainly demands further investigation perhaps using an ensemble of observational data sources.

**Comment 9.**
Figure 4b: Based on L362, I would expect the red-dashed E-PI trend to be at 0.09 m/s/year here, not 0.12 m/s/year at is appears to be

Thank you for spotting our error. This has been corrected.

**Comment 10.**
Figure 4: I think it could be interesting to see quantile changes vs E-PI, but I leave it up to the authors to decide if they think it adds to the paper and if they want to add it.

Great comment. It has been our experience with colleagues that these types of quantile plots are already challenging to interpret when seeing them for the first time. We therefore choose not to add more complexity.

**Comment 11.**
L433-L434: It is worth specifying in the methods whether PI in the simulations is computed a priori (e.g., using pyPI at the initial time) or in situ (e.g., in Bryan and Rotunno 2009). If, as I seem to understand, the computation of PI occurs at the initial time based on the temperature profile, this interesting discrepancy may be due to a

mismatch between the E-PI-calculated outflow temperature and the realized outflow temperature of the storm.

Thank you for this insight. You are correct that we used pyPI at initial time. This temporal inconsistency and its possible implications are now made clear throughout the paper.

**Comment 12.**
L441-L453: I don't know whether saying that the differences between the experiments are "small" is very useful, given the fact that the ensembles perturb numerical and physical parameters to which the TC intensity is highly sensitive. Since it is physics that is perturbed in the ensemble, not initial conditions, and we don't really expect physics to change with climate, wouldn't it make sense to simply take the central pressure difference between each corresponding ensemble member in different warming scenarios (i.e. same physics in present day and end of century)? Then the distribution of pressure differences could be used to produce a box plot of central pressure change and to verify whether change is statistically different from zero. I think this may yield similar results to the Wilcoxon test you are performing here, so please only add this if you feel that it would improve the paper. Disregard otherwise.

Thank you for this comment. Your suggestion to analyze the pressure differences (future – current climate) for each ensemble member is a good one and is perhaps a more elegant presentation of the results. But we are confident that it should lead to similar conclusions to our analysis of the ensemble spread in pressures for each experiment already shown in Fig. 6b. We therefore choose to keep our original Wilcoxon tests and Fig. 6b. However, we do remove our statement that differences between experiments are small.

**Comment 13.**
L505-L508 and elsewhere: This is one instance of Major comment 1.1. For an imaginary completely fixed temperature profile, if SST increases, thermodynamic efficiency will also increase, but the biggest contribution to PI by far will come from thermodynamic disequilibrium, not efficiency. Hence this interpretation that changes in intensity are explained by the changes in efficiency is incomplete.

This specific example of our misinterpretation has been corrected as suggested. So too have other related misinterpretations in the manuscript.

* Final note: I really appreciated reading this paper and I hope it can be published!
I cite a few papers in this revision, including some of my own, which was meant to provide examples. Please do not feel the need to cite them.
Raphaël Rousseau-Rizzi

Thank you. We found your own series of papers and Gilford et al. (2017) to be fantastic and very relevant to our study.

**References**

Y. P. Bu, R. G. Fovell, and K. L. Corbosiero. Influence of cloud–radiative forcing on tropical cyclone structure. Journal of the Atmospheric Sciences, 71(5):1644–1662, 2014.

S. Garner. The relationship between hurricane potential intensity and cape. Journal of the Atmospheric Sciences, 72(1):141–163, 2015.

D. M. Gilford, S. Solomon, and K. A. Emanuel. On the seasonal cycles of tropical cyclone potential intensity. Journal of Climate, 30(16):6085–6096, 2017.

R. Rousseau-Rizzi and K. Emanuel. A weak temperature gradient framework to quantify the causes of potential intensity variability in the tropics. Journal of Climate, 34(21): 8669–8682, 2021.

R. Rousseau-Rizzi, R. Rotunno, and G. Bryan. A thermodynamic perspective on steadystate tropical cyclones. Journal of the Atmospheric Sciences, 78(2):583–593, 2021.

R. Rousseau-Rizzi, T. M. Merlis, and N. Jeevanjee. The connection between carnot and cape formulations of tc potential intensity. Journal of Climate, 35(3):941–954, 2022.